# Osteoblastic Swedish mutant APP expedites brain deficits by inducing endoplasmic reticulum stress-driven senescence

Jin-Xiu Pan[1,2,3], Dong Sun[1,3], Daehoon Lee[1,2], Lei Xiong[1,2], Xiao Ren[1], Hao-han Guo[1], Ling-Ling Yao[1], Yuyi Lu[1], Caroline Jung[1] & Wen-Cheng Xiong [1,2✉]

Patients with Alzheimer's disease (AD) often have osteoporosis or osteopenia. However, their direct link and relationship remain largely unclear. Previous studies have detected osteoporotic deficits in young adult $Tg2576$ and $TgAPP_{swe}^{OCN}$ mice, which express $APP_{swe}$ (Swedish mutant) ubiquitously and selectively in osteoblast (OB)-lineage cells. This raises the question, whether osteoblastic $APP_{swe}$ contributes to AD development. Here, we provide evidence that $TgAPP_{swe}^{OCN}$ mice also exhibit AD-relevant brain pathologies and behavior phenotypes. Some brain pathologies include age-dependent and regional-selective increases in glial activation and pro-inflammatory cytokines, which are accompanied by behavioral phenotypes such as anxiety, depression, and altered learning and memory. Further cellular studies suggest that $APP_{swe}$, but not $APP_{wt}$ or $APP_{lon}$ (London mutant), in OB-lineage cells induces endoplasmic reticulum-stress driven senescence, driving systemic and cortex inflammation as well as behavioral changes in 6-month-old $TgAPP_{swe}^{OCN}$ mice. These results therefore reveal an unrecognized function of osteoblastic $APP_{swe}$ to brain axis in AD development.

[1] Department of Neurosciences, School of Medicine, Case Western Reserve University, Cleveland, OH, USA. [2] Louis Stokes Cleveland Veterans Affairs Medical Center, Cleveland, OH, USA. [3] These authors contributed equally: Jin-Xiu Pan, Dong Sun. ✉email: Wen-Cheng.Xiong@case.edu

Alzheimer's disease (AD) is the most common form of dementia. It is pathologically characterized by cortical and cerebrovascular β-amyloid (Aβ) plaques, phosphor-tau containing neurofibrillary tangles, reactive glial cell (astrocyte and microglial cell)-associated chronic brain inflammation, and neuron-loss[1,2]. Interestingly, in addition to brain pathologies, patients with AD, both early and late onset, often have osteopenia or osteoporosis[3–10], a condition characterized by the loss of bone-mass or bone mineral density (BMD) with micro-architectural deterioration of bone tissue, and a higher rate of hip fracture. However, little is known regarding the underlying mechanisms of AD association with bone loss.

A growing list of genetic risk genes has been identified in patients with early onset and late onset AD. Intriguingly, many of the AD risk genes, such as TREM2 (triggering receptors expressed on myeloid cells-2) and PYK2, are highly expressed in immune cells and bone cells, and encode proteins that regulate not only neuron synaptic functions, but also immune responses and bone homeostasis[11–15]. APOE, another AD risk gene, is also identified as a risk factor for osteoporosis[16–18]. Among the various risk genes for AD development, we chose Swedish mutant APP (APP_{swe}) to address the question regarding AD association with bone loss for the following reasons. The Swedish mutations in the APP gene are initially identified in patients with early-onset (EO) AD, which promote the generation of Aβ by favoring its pro-teolytic cleavage performed by β- and γ-secretases[19–21]. Much research has focused on the impacts of Aβ on the brain, even though APP or APP_{swe} is known to be expressed not only in the brain, but also in periphery tissues, including osteoblast (OB)-lineage cells[22,23]. Although APP_{swe} is only detected in a small fraction of AD patients, it is commonly used to generate AD animal models, such as Tg2576 and 5XFAD[24,25]. APP_{swe} in these animal models (in particularly Tg2576) is expressed ubiquitously, in both the brain and periphery tissues, including OB cells[22,23]. While investigating the phenotypes of these APP_{swe}–based animal models have provided valuable insights into Aβ brain pathology and impairments in mouse cognitive functions, the function of APP_{swe} in peripheral tissues, such as OBs, remains poorly understood. Previous examinations of bone structures in Tg2576 mice have identified early-onset osteoporotic deficits, months before any brain-pathologic defect that was detected[22,23]. Knocking out App (in APP[−/−] mice), or selective expressing APPswe in osteocalcin (OCN) promoter driven Cre (OCN-Cre)[+] OB-lineage cells (in TgAPP_{swe}^{OCN} mice) recapitulates the osteoporotic defects in Tg2576 mice[23,26]. These observations raise an interesting question, could problems in the bone cells conversely contribute to AD pathology in the brain?

Here, we provide evidence that TgAPP_{swe}^{OCN} mice express APP_{swe} largely in the OB-lineage cells, with little to weak expression in the dorsal dentate gyrus (dDG) of the hippo-campus. These mice develop age-dependent [starting at 6-month-old (MO)] and brain-region selective pathologies, and exhibit anxiety- and depression-like behaviors, as well as altered cognitive functions. While these mice at 6-MO showed brain-pathy (including glial activations and elevated pro-inflammatory cytokines) largely in the cortex, these mice at 12-MO showed brain-pathy mainly detected in the hippocampus. Further mechanistic studies demonstrate that APP_{swe}, but not APP_{wt} or APP_{lon} (London mutant), in OB-lineage cells increases endo-plasmic reticulum (ER)-stress, senescence, and SASPs (senes-cence associated secretory phenotypes). Inhibition of ER-stress abolishes APP_{swe}-induced senescence, and suppression of senescence diminishes brain and behavioral phenotypes in 6-MO TgAPP_{swe}^{OCN} mice. Taken together, these observations suggest that APP_{swe} in OB-lineage cells contributes to the brain-region selective inflammation and glial activation and induces anxiety-

and depression-like behaviors in age-dependent manner, which are largely due to elevated OB-senescence, SASPs, and systemic inflammation. These results thus uncover a link between APP_{swe} in the OB-lineage cells and AD development.

## Results

**Selective APP_{swe} expression in OB-lineage cells in TgAPP_{swe}^{OCN} mice.** To investigate osteoblastic APP_{swe}'s function in AD development, we took advantage of TgAPP_{swe}^{OCN} mice, in which human APP_{swe} expression in LSL-hAPP_{swe} mice depends on the removal of LSL by the OCN-Cre (Fig. 1a)[23]. Although OCN-Cre mice express Cre primarily in mature/adult OB-lineage cells[27,28], our recent study showed Cre activity in neurons of dDG hippocampus, olfactory bulb, and cerebellum[29]. Thus, it is important to verify APP_{swe}'s expression in bone cells and brain tissues of TgAPP_{swe}^{OCN} mice. Notice that the hAPP_{swe} protein was detected in the OB-lineage BMSCs (bone marrow stromal cells), but not in the hippocampus or cortex of the TgAPP_{swe}^{OCN} mice (6-MO) (Fig. 1b, c). We then asked if this is due to hAP-P_{swe}'s cleavage (to produce Aβ_{40} or Aβ_{42}) in the brain tissues. ELISA measuring human Aβ_{40} and Aβ_{42} levels showed little-to-no Aβ increase in the hippocampus, cortex, or serum samples (Fig. 1d, e); but slight increases of both Aβ_{40} and Aβ_{42} in the OB-lineage cells, of TgAPP_{swe}^{OCN} mice (6-MO), as well as in the brain tissues and serum samples of 6-MO Tg2576 mice (Fig. 1d, e). These results eliminate the possibility of β- and γ-cleavages of hAPP_{swe} in the brain of 6-MO TgAPP_{swe}^{OCN} mice, suggesting little hAPP_{swe} expression in the mutant brain at this age. We further tested this view by RT-PCR analysis of hAPP_{swe}'s tran-scripts in the mutant mice. Using specific primers for human APP, a weak hAPP_{swe} expression (~1.5 fold over control) was detected in the TgAPP_{swe}^{OCN} brain regions (e.g., hippocampus, olfactory bulb, and cerebellum) where OCN-Cre is expressed[29], but not in the OCN-Cre negative cortex (Fig. 1f). Notice that the hAPP_{swe}'s transcripts were much more abundant in the BMSCs (~70 fold over control) than in the brain (Fig. 1f), implying a much weaker Cre activity in neurons than in OB-lineage cells of the OCN-Cre mice. This viewpoint is consistent with the RT-PCR findings that Cre is expressed largely in the OB-lineage cells (~128 fold over control), weakly (~18 fold over control) in the hippo-campus, and undetectable in the cortex of OCN-Cre mice (Fig. 1g). Taken together, these results suggest that the hAPP_{swe} is highly expressed in OCN-Cre[+] OB-lineage cells, but little to weakly expressed in the OCN-Cre[+] dDG, olfactory bulb, and cerebellum neurons, of TgAPP_{swe}^{OCN} mice.

**Age-dependent and brain region-selective elevations in reactive astrocytes, microglial cells, and inflammatory cytokines, and an impairment in DG neurogenesis in TgAPP_{swe}^{OCN} mice.** We then addressed whether TgAPP_{swe}^{OCN} mice exhibit any brain pathology that is similar to those of APP_{swe}-based AD animal models (e.g., Tg2576)[24,25,32–34], by performing the following studies.

First, we measured both Aβ_{40} and Aβ_{42} levels in the bone cells and brain tissues of TgAPP_{swe}^{OCN} mice at ages of not only 6-MO, but also 12-MO. Although little Aβ_{40} or Aβ_{42} levels were detected in 6-MO TgAPP_{swe}^{OCN} cortex and hippocampus (Fig. 1d, e), Aβ_{42}, but not Aβ_{40}, was slightly elevated in 12-MO TgAPP_{swe}^{OCN} hippocampus, but not cortex nor serum samples (Supplementary Fig. 1a, b). Additionally, little to no Aβ plaque was detected in 12-MO TgAPP_{swe}^{OCN} bone and brain sections, in contrast from brain sections from 5XFAD mice (4.5 MO) (Supplementary Fig. 1c–e). These findings support the view for a weak hAPP_{swe}/Aβ_{42} expression in 12-MO TgAPP_{swe}^{OCN} hippocampal DG neurons.

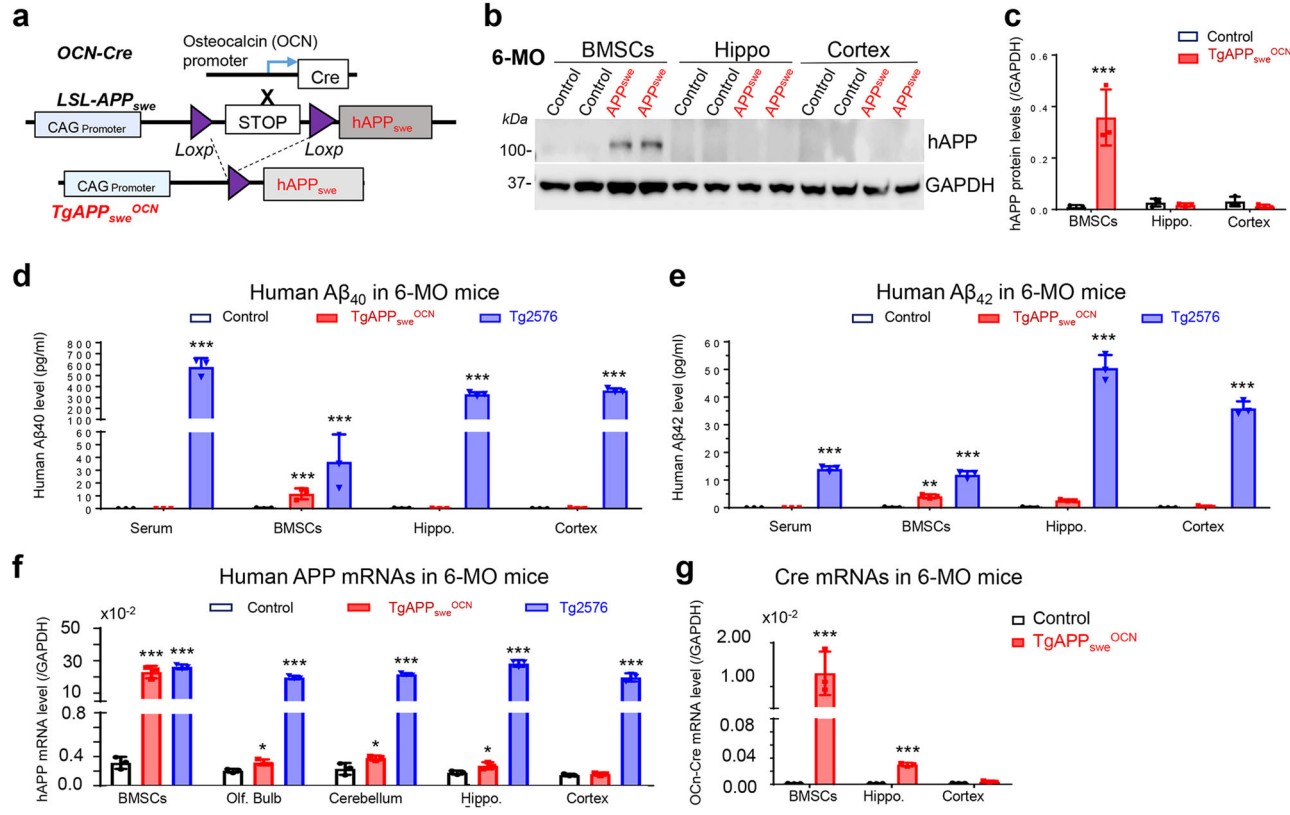

**Fig. 1 Specific expression of APP_swe in OB-lineage cells in *TgAPP_swe^OCN* mice. a** Illustration of the transgene and generation of the conditional transgenic mice selectively expressing human *APP_swe* in an OCN-Cre dependent manner. **b**, **c** Western blot analysis of human APP (hAPP) protein levels in BMSCs, hippocampus, and cortex of 6-MO control (*LSL-APP_swe*) and *TgAPP_swe^OCN* mice. **b** Representative blots; and **c** quantification. **d**, **e** ELISA analyses of human Aβ$_{40}$(**d**) and Aβ$_{42}$(**e**) levels in serum, BMSCs (50 μg in total protein), and brain homogenates including hippocampus and cortex (300 μg total protein) from 6-MO control, *TgAPP_swe^OCN*, and *Tg2576* mice. **f** RT-PCR analysis of *hAPP* gene expression in BMSCs, olfactory bulb, cerebellum, hippocampus, and cortex of 6-MO control and *TgAPP_swe^OCN* mice. **g** RT-PCR analysis of *Cre* expression in BMSCs, hippocampus, and cortex of 6-MO control (*LSL-APP_swe*) and *TgAPP_swe^OCN* mice. All data were presented as mean ± SD. *$p < 0.05$, **$p < 0.01$, ***$p < 0.001$ ($n = 3$ mice). Mann–Whitney $U$ test was used in **c** and **g**, and one-way ANOVA followed by Tukey post hoc test was used in **d**–**f**.

Second, we examined neuronal distribution patterns and densities in the cortex and hippocampus of *TgAPP_swe^OCN* mice (at age of ~7-MO) by conducting co-immunostaining analysis using antibodies against NeuN (a marker for all neurons) and Ctip2 (a marker for Layer V–VI neurons in the cortex and neurons in CA1-2 and DG). Little change in the NeuN$^+$ and Ctip2$^+$ neuron distribution patterns and densities was detected in *TgAPP_swe^OCN* brains (Supplementary Fig. 2).

Third, we assessed the morphologies and densities of glial cells, including Olig2$^+$ oligodendrocytes, S100β$^+$ ependymal cells, GFAP$^+$ astrocytes, and IBA1$^+$ microglial cells, in the brain sections of control (*LSL-APP_swe*) and *TgAPP_swe^OCN* mice. The Olig2$^+$ oligodendrocytes and S100β$^+$ ependymal cells appeared to be unchanged in the *TgAPP_swe^OCN* cortex or brain (Supplementary Fig. 3). Intriguingly, both GFAP$^+$ astrocytes and IBA1$^+$ microglial cells were increased in 6-MO *TgAPP_swe^OCN* cortex, particularly in layers I–III, but not in hippocampus (Fig. 2a–d), suggesting a brain region-selective activation of these glial cells. This view was further verified through a Western blot analysis, which showed increased GFAP and IBA1 protein levels in 6-MO *TgAPP_swe^OCN* cortex, but not in hippocampus (Fig. 2e, f). Because glial cell activation is often associated with increased inflammation[30,31], we examined expressions of inflammation associated cytokines (e.g., *Il1b*, *Il6*, *Il10*, and *Tnfa*), growth factors (e.g., *Tgfb1* and *Csf2*), and proteinase (e.g., *Mmp3*) in both the cortex and hippocampus of control and *TgAPP_swe^OCN* mice (at 6-MO) using RT-PCR analysis. The transcripts of *Il1b*, *Il10*, *Tnfa*,

and *Mmp3* were all increased in *TgAPP_swe^OCN* cortex, but not in hippocampus (Fig. 2g, h), supporting the view of cortex as a vulnerable brain region in 6-MO *TgAPP_swe^OCN* mice.

Fourth, we found that the glial activation and inflammatory phenotypes in *TgAPP_swe^OCN* mice were not only brain-region selective, but also age-dependent. Whereas the cortex displayed the glial activation/inflammation in 6-MO *TgAPP_swe^OCN* mice, these phenotypes were not detected in 3-MO *TgAPP_swe^OCN* (Supplementary Fig. 4), but evidently more obvious in 12-MO *TgAPP_swe^OCN* hippocampus than the cortex (Supplementary Fig. 5), suggesting age-dependent changes in the brain-region selectivity of the glial activation/inflammation phenotypes.

Finally, we examined adult neurogenesis in hippocampal DG (dentate gyrus), which is also impaired in AD animal models[32]. EdU was injected into the mice ~12 h before sacrifice to label proliferative neural stem cells (NSCs). Hippocampal sections were co-immunostained EdU with antibodies against DCX (doublecortin) (a marker for newborn neurons derived from NSCs). While *TgAPP_swe^OCN* mice at 3-MO showed no difference in EdU$^+$ and DCX$^+$ cell densities compared to the controls, *TgAPP_swe^OCN* mice at 6-MO displayed significant reductions in EdU$^+$ and DCX$^+$ cell densities at both dorsal and ventral DG (Supplementary Fig. 6), demonstrating an age-dependent impairment in the hippocampal DG neurogenesis of *TgAPP_swe^OCN* mice, similar to that described in AD animal models[32].

In aggregate, *TgAPP_swe^OCN* mice (starting at 6-MO) exhibit partial AD relevant brain pathologies, which include increased

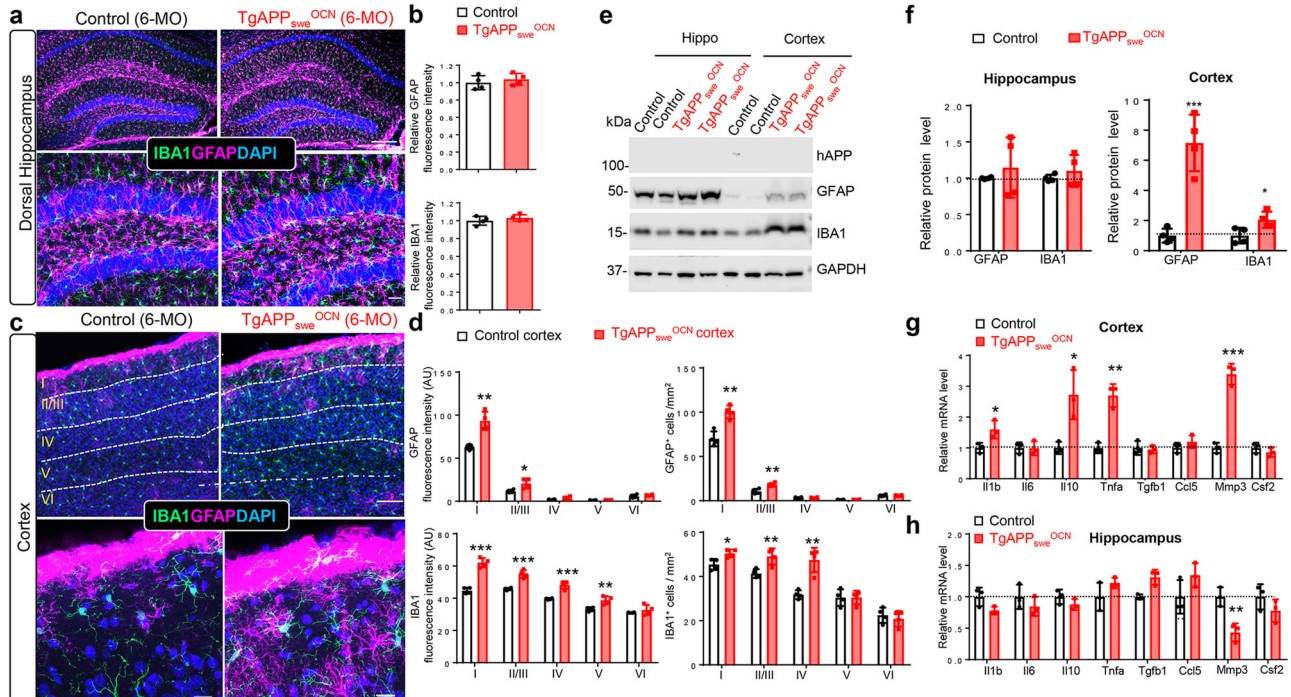

**Fig. 2 Elevated reactive astrocytes, microglial cells, and inflammatory cytokines in 6-MO *TgAPP_{swe}^{OCN}* cortex, but not hippocampus. a** Representative images of co-immunostaining with IBA1 (green), GFAP (magenta), and DAPI (blue) of hippocampal sections from 6-MO control (*LSL-APP_{swe}*) and *TgAPP_{swe}^{OCN}* mice. Scale bars: 200 μm (upper) and 20 μm (lower). **b** Quantification of data in **a**. **c** Representative images of co-immunostaining with IBA1 (green), GFAP (magenta), and DAPI (blue) of cortex sections from 6-MO control (*LSL-APP_{swe}*) and *TgAPP_{swe}^{OCN}* mice. Scale bars: 100 μm (upper) and 20 μm (lower). **d** Quantification of data in **c**. **e** Representative Western blots using antibodies against hAPP, GFAP, and IBA1 in homogenates of cortex and hippocampus of control and *TgAPP_{swe}^{OCN}* mice. GAPDH was used as a loading control. **f** Quantification of the data in **e**. **g**–**h** Real-time PCR (RT-PCR) analysis of indicated gene expressions in 6-MO control (*LSL-APP_{swe}*) and *TgAPP_{swe}^{OCN}* cortex (**g**) and hippocampus (**h**). All quantification data were presented as mean ± SD ($n = 3$–4). *$p < 0.05$, **$p < 0.01$, ***$p < 0.001$. Student's $t$ test was used in **b**, **d**, and **f**–**h**.

reactive astrocytes, microglial cells, and inflammatory cytokines in the cortex (at 6-MO)/hippocampus (at 12-MO), impaired DG neurogenesis, and elevated Aβ₄₂ in 12-MO hippocampus.

**Age-dependent anxiety- and depression-like behaviors in *TgAPP_{swe}^{OCN}* mice.** Glial activation, brain inflammation, and decreased DG neurogenesis are often associated with depression- and/or anxiety-like behaviors[33–38]. We thus subjected *TgAPP_{swe}^{OCN}* and control mice to an open field test (OFT) for evaluation of *TgAPP_{swe}^{OCN}* mice's anxiety and locomotor activity. *TgAPP_{swe}^{OCN}* mice at 6- and 12-MO, but not 3-MO, showed reduced center duration time but comparable total distance traveled to the controls (Fig. 3a, b and Supplementary Fig. 7a, b), suggesting a reduced exploratory, but not locomotor, activity, in the mutant mice, and implicating anxiety and/or depression. We further examined their behaviors using elevated plus maze test (EPMT) and light/dark transition test (LDT), both tests widely used to assess anxiety-related behavior in mouse models[39,40]. Indeed, *TgAPP_{swe}^{OCN}* mice, again at 6- and 12-MO, but not 3-MO, showed decreased open arm duration time and entries by EPMT (Fig. 3c, d and Supplementary Fig. 7c, d), and reduced time in light box room in the LDT (Fig. 3e and Supplementary Fig. 7e), supporting the view for anxiety-like behaviors. We then assessed their depression-like behaviors using tail suspension test (TST), force swimming test (FST), and sucrose preference test (SPT). *TgAPP_{swe}^{OCN}* mice (6- and 12-MO, but not 3-MO) appeared to be depressed, exhibiting increased immobility times in both TST (Fig. 3f and Supplementary Fig. 7f) and FST (Fig. 3g and Supplementary Fig. 7g) and reduced sucrose preference (Fig. 3h and Supplementary

Fig. 7h). Together, these results suggest that *TgAPP_{swe}^{OCN}* mice experience age-dependent (starting at 6-MO) anxiety- and depression-like behaviors.

Since hAPP_{swe} is weakly expressed in dDG neurons in *TgAPP_{swe}^{OCN}* hippocampus (Fig. 1f), we wondered whether such a weak dDG expression of APP_{swe} could induce similar behavior phenotypes to that in *TgAPP_{swe}^{OCN}* mice. The AAV-CamkII-Cre (Cre under the control of CamkII promotor for excitatory neuron expression) and AAV-CamkII-GFP (as control) were specifically injected into the dDGs of both sides of the hippocampus in *LSL-APP_{swe}* mice (at age of 4-MO); and mice at 6-MO were subjected to the behavior tests (Supplementary Fig. 8a). While dDG neurons in *LSL-APP_{swe}* mice were successfully infected with the viruses (indicated by the GFP, hAPP_{swe} expression, and Aβ₄₂ increase) (Supplementary Fig. 8b–g), little to no differences in behavior tests using EMPT, LDT, TST, FST, and SPT were detected between Cre and GFP virus injected mice (Supplementary Fig. 8h–l), unlike the *TgAPP_{swe}^{OCN}* mice. These results thus implicate that the anxiety- or depression-like behaviors in 6-MO *TgAPP_{swe}^{OCN}* mice are in large due to APP_{swe}'s expression in OB-lineage cells, but not dDG neurons.

**Age-dependent alterations in spatial learning and memory in *TgAPP_{swe}^{OCN}* mice.** Although anxiety- and depression-like behaviors are present in AD animal models (e.g., Tg2576 and 5XFAD)[41–43] and AD patients[44–46], a key AD relevant functional deficit is the age-dependent cognition decline[47,48]. Therefore, we subjected *TgAPP_{swe}^{OCN}* and control (*LSL-APP_{swe}*) mice to the Morris water maze (MWM) test (to access mouse spatial learning and memory function)[49], and the novel object recognition (NOR)

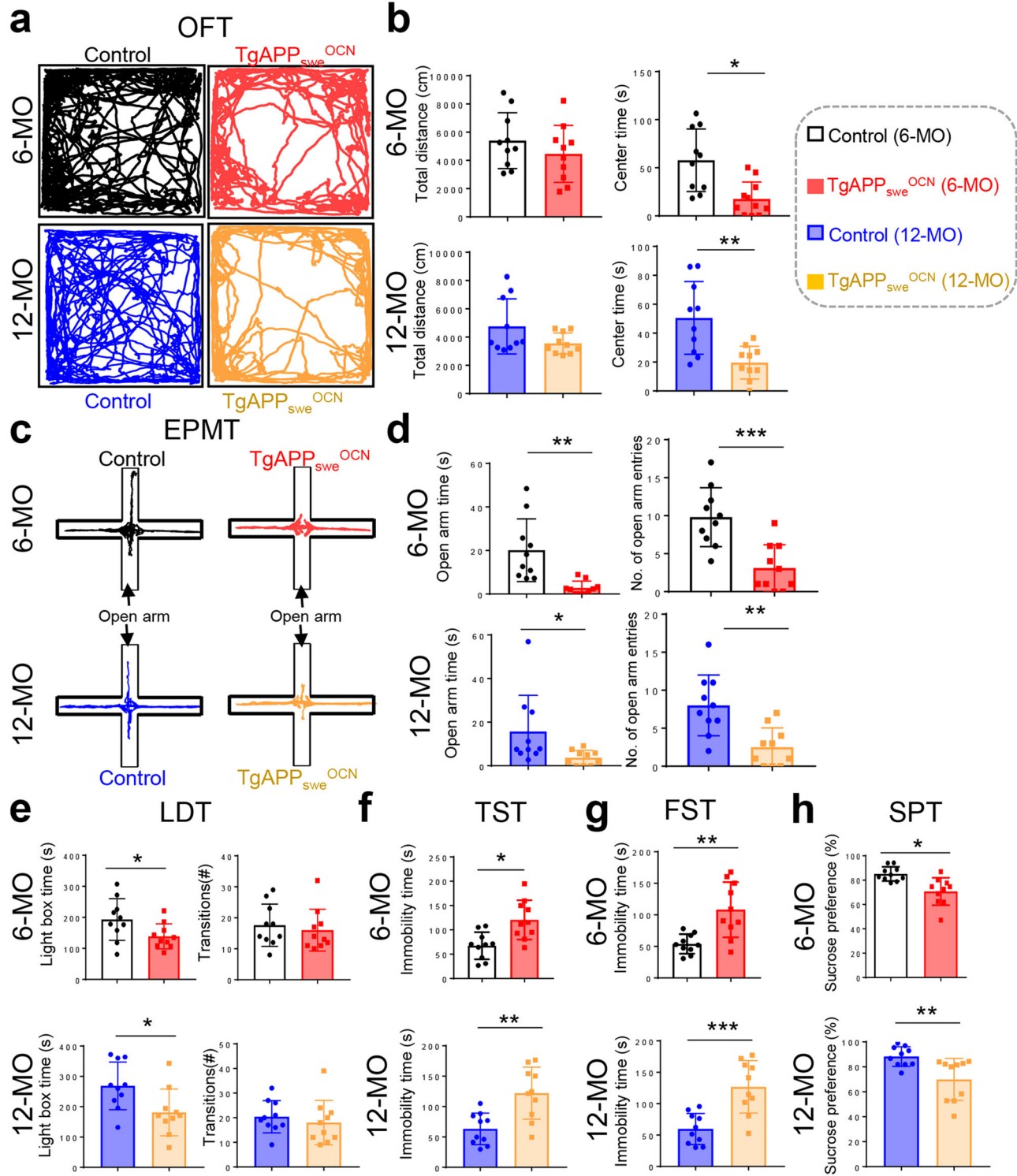

**Fig. 3 Age-dependent anxiety- and depression-like behaviors in *TgAPP_swe^OCN* mice. a, b** OFT: Representative tracing images (**a**), and quantifications of total distance and center duration time (**b**) were shown. **c, d** EPMT: Representative tracing images (**c**), and quantifications of open arm duration time and entries (**d**) were shown. **e** LDT: Quantifications of the time spent in the light room and the number of transitions into the light room. **f** TST, **g** FST, and **h** SPT. In all these behavior tests, 6-MO and 12-MO control (*LSL-APP_swe*) and *TgAPP_swe^OCN* mice (males) were examined. All quantification data were shown as mean ± SD ($n = 10$ mice). *$p < 0.05$, **$p < 0.01$, ***$p < 0.001$, Student's $t$ test.

test (to evaluate mouse recognition memory)[50,51]. Interestingly, age-dependent changes in both MWM and NOR tests were detected in *TgAPP_swe^OCN* mice. No obvious difference in MWM or NOR task performance was observed between *TgAPP_swe^OCN* and control mice at 3-MO (Fig. 4a–c). Un-expectedly, at 6-MO,

*TgAPP_swe^OCN* mice exhibited faster learning and better long-term memory in MWM (Fig. 4d, e), but no obvious difference in NOR task performance (Fig. 4f), compared to the age-matched controls, suggesting an improvement in spatial learning and memory in 6-MO *TgAPP_swe^OCN* mice. Interestingly, at ~12-MO,

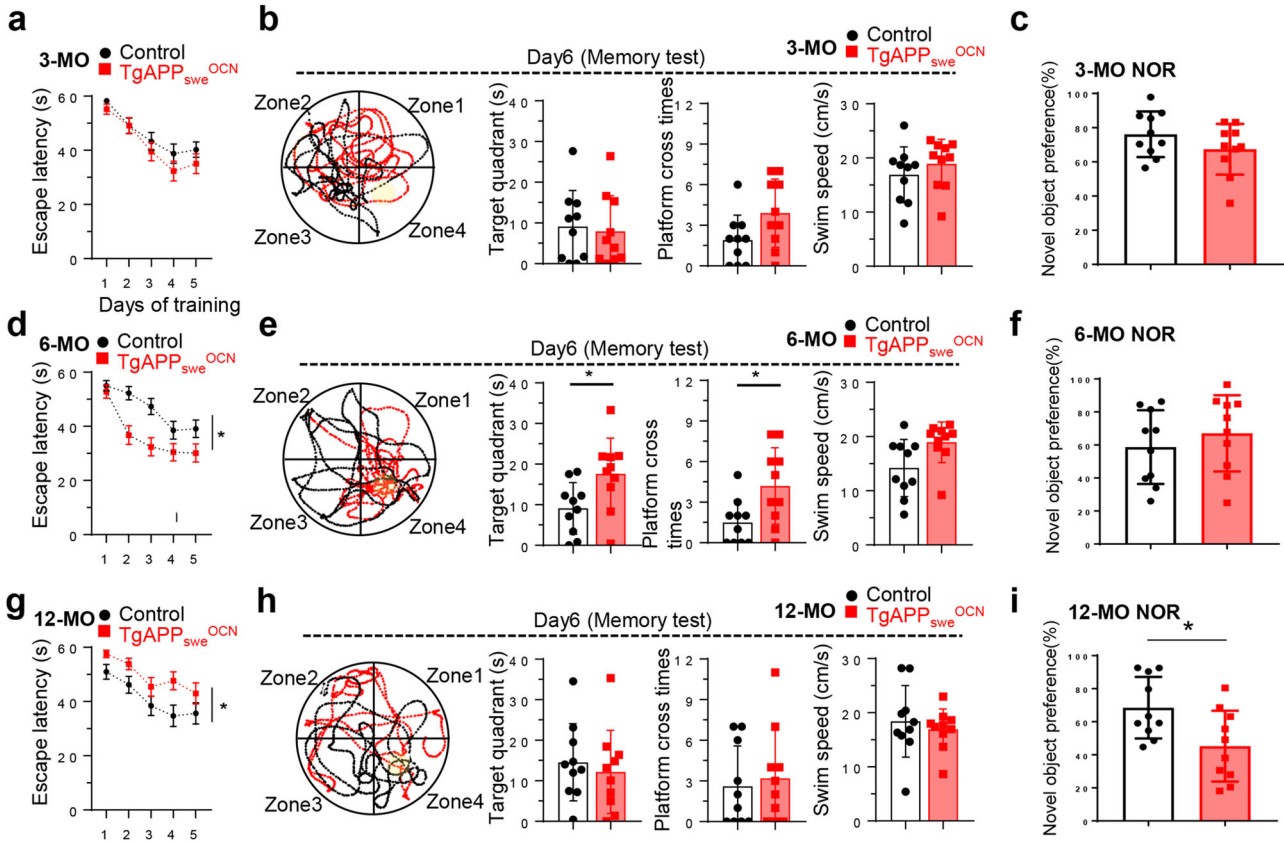

**Fig. 4 Age-dependent alterations in spatial learning and memory in $TgAPP_{swe}^{OCN}$ mice. a–c** 3-MO control ($LSL-APP_{swe}$) and $TgAPP_{swe}^{OCN}$ male mice were subject to Morris water maze (MWM) (**a**, **b**) and Novel Object Recognition (NOR) (**c**) tests. **d–f** 6-MO control ($LSL-APP_{swe}$) and $TgAPP_{swe}^{OCN}$ male mice were subject to MWM (**d**, **e**) and NOR (**f**) tests. **g–i** 12-MO control ($LSL-APP_{swe}$) and $TgAPP_{swe}^{OCN}$ male mice were subject to MWM (**g**, **h**) and NOR (**i**) tests. In MWM tests, the latencies to reach the hidden platform during the training period were showed in **a**, **d**, and **g**; and the representative tracing images and quantification of time spent in target quadrant, platform crossing time, and swim speed were shown in **b**, **e**, and **h**. In NOR tests, the time spent with novel object per total time with both objects as the novel object preference was quantified, shown in **c**, **f**, and **i**. All values were presented as mean ± SD ($n = 10$ mice). *$p < 0.05$, one-way ANOVA followed by Tukey post hoc test was used in **a**, **d**, and **g**, and Student's $t$ test was used in **b**, **c**, **e**, **f**, **h**, and **i**.

impairments in both MWM and NOR tasks were detected in $TgAPP_{swe}^{OCN}$ mice (Fig. 4g–i). These results are intriguing, demonstrating age-dependent changes in spatial and novel object learning and memory of $TgAPP_{swe}^{OCN}$ mice, in line with their age-dependent changes in the brain-region selectivity of the glial activation/inflammation.

**Increased senescence and SASPs in $APP_{swe}$+ OB-lineage cells**. To investigate if and how $APP_{swe}$ in OCN-Cre+ OB-lineage cells gives rise to the brain and behavior phenotypes, we purified OCN-Cre+ BMSCs (marked by tdTomato+, believed to be OB progenitors[28]) from both 6-MO control (*OCN-Cre; Ai9*) and $TgAPP_{swe}^{OCN}$; *Ai9* mice using fluorescence-activated cell sorting (FACS), and then subjected them to RNA-seq analysis (Fig. 5a). 917 up- and 1825 down-regulated genes were identified in $APP_{swe}$+ OB progenitors (Fig. 5b). Among these genes, 154 up- and 269 down-regulated genes encode secreted proteins (Fig. 5b). Interestingly, GO analysis showed that most up-regulated genes are involved in inflammatory response, cytokine production, and cytokine/chemokine-mediated signaling pathways; and most down-regulated genes are implicated in cell cycle, cell proliferation, and bone mineralization (Fig. 5c). Further heat map analysis illustrated the up- and down-regulated genes for bone mass regulators, cytokines and chemokines, AD risk genes, and growth factors critical for neurogenesis (Fig. 5d). Some of these up/down regulated genes were verified by RT-PCR analyses (Fig. 5e).

Notice that transcription factors (e.g., *Sp7*, *Nfatc1*, *Satb2*, *Spp1*, *Sparc*, and *Jun*) for OB-lineage cell proliferation, differentiation, or mineralization, and bone-mass regulators (e.g., *Bmp2*, *Ihh*, *Lrp4*, and *Ctnnb1*) were down regulated in $APP_{swe}$+ OB progenitors (Fig. 5d, e), in line with our previous report[23]; and some of the AD-risk genes including *Trem2*, *ApoE*, *Cd33*, *Sorl1*, *Vps35*, *Ptk2b*, and *Psen1/2* were altered in $APP_{swe}$+ OB progenitors, and growth factors including brain-derived neurotrophic factor (*Bdnf*) and insulin-like growth factor 1 (*Igf1*) were decreased in $APP_{swe}$+ OB progenitors (Fig. 5d, e).

Interestingly, the increased cytokines and chemokines exhibits features of SASPs[52,53]. We thus asked if $APP_{swe}$+ OB-progenitors undergo senescence. Indeed, senescence marker proteins, such as SA-β-gal (senescence associated β-galactosidase), P16$^{Ink4a}$, P21, and P53[54,55], were all elevated in $APP_{swe}$+ OB-progenitors (derived from 3- and 6-MO mice) (Fig. 6), suggesting that $APP_{swe}$ induces OB-senescence. We also detected reductions in tdTomato+ (Td+) or OCN-Cre+ OB progenitors (Supplementary Fig. 9a, b) and in EdU+ proliferative cells in OB-progenitor cultures from $TgAPP_{swe}^{OCN}$; *Ai9* mice (Supplementary Fig. 9c–e), indicating a growth arrest of these cells, another feature of cellular senescence in $APP_{swe}$+ OB-progenitors.

Finally, we wondered whether other tissues in $TgAPP_{swe}^{OCN}$ mice develop senescence-like phenotypes. The mRNAs from various tissues [including cortex, hippocampus, TA (Tibialis anterior) muscles, kidneys, and livers] of 6-MO control and $TgAPP_{swe}^{OCN}$

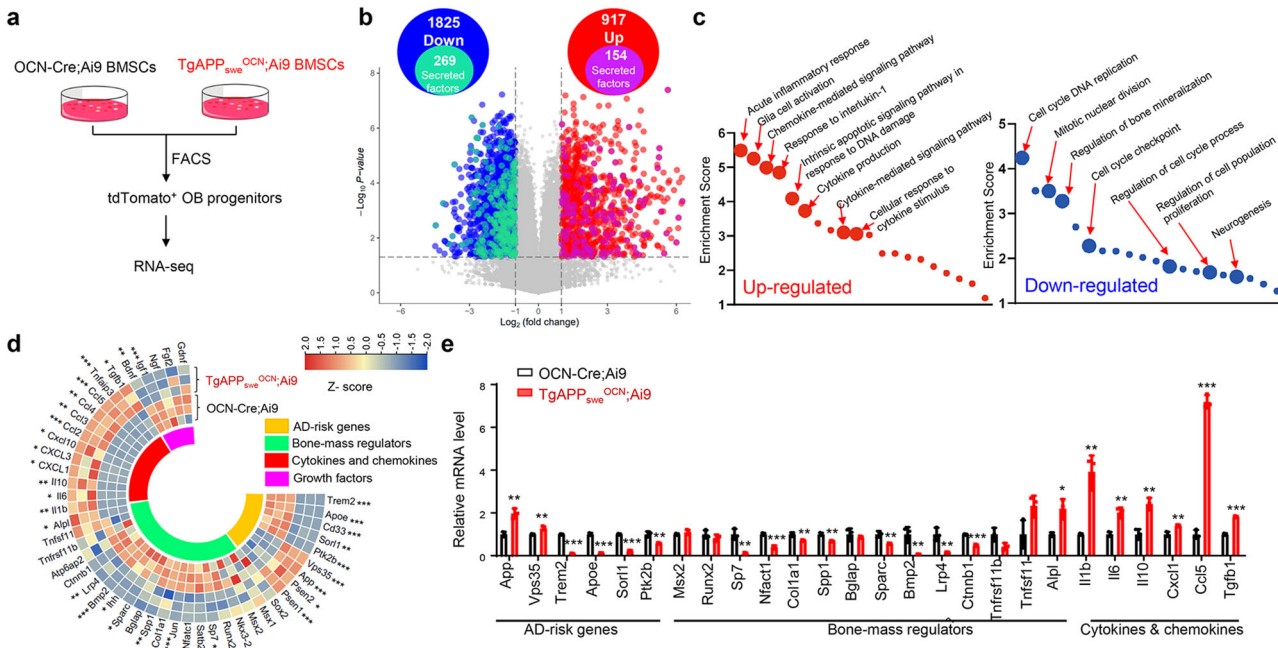

**Fig. 5 Increased cytokines and chemokines in APP_swe⁺ OB-progenitor cells. a** Schematic of purification and RNA-seq of Tdtomato⁺ (Td⁺) OB progenitors from control (*OCN-Cre; Ai9*) and *TgAPP_swe^OCN; Ai9* mice. **b–d** Volcano plots (**b**), GO analysis of up/down-regulated genes (**c**), and heat map (**d**) of differentially expressed genes identified by RNA-seq. **e** RT-PCR analysis of AD risk gene *App, Vps35, Trem2, Apoe, Ptk2b,* and *Sorl1*; bone-mass regulator *Sp7, Nfatc1, Col1a1, Spp1, Sparc, Bmp2, Lrp4,* and *Ctnnb1*; cytokine *Il1b, Il6,* and *Il10*; chemokine *Ccl5* and *Cxcl1*, growth factor *Tgfb1* gene expression in purified Td⁺ OB progenitors from 6-MO control (*OCN-Cre; Ai9*) and *TgAPP_swe^OCN; Ai9* mice. All values were presented as mean ± SD (*n* = 3 mice). *$p < 0.05$, **$p < 0.01$, and ***$p < 0.001$, by Mann–Whitney *U* test.

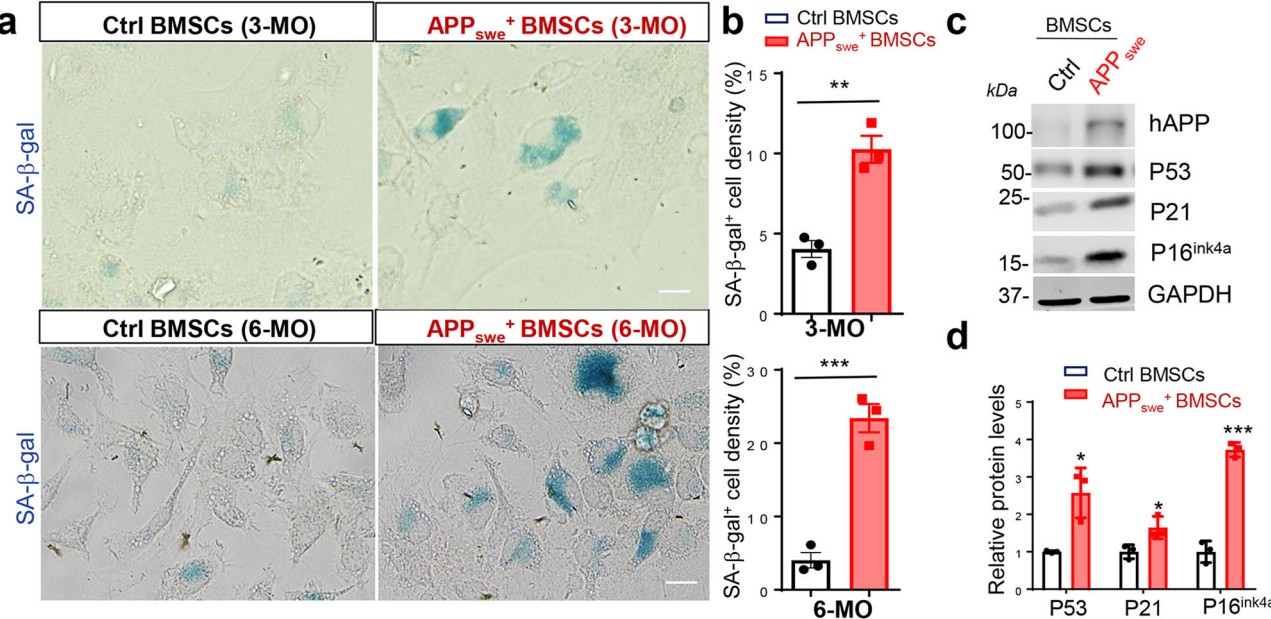

**Fig. 6 Increased cellular senescence in APP_swe⁺ OB-lineage cells. a** SA-β-gal staining of 3-MO and 6-MO BMSCs from control (*LSL-APP_swe*) and *TgAPP_swe^OCN* mice. Scale bar, 20 μm. **b** Quantification of SA-β-gal⁺ cell densities (mean ± SD; *n* = 3 independent experiments). **$p < 0.01$, ***$p < 0.001$. **c** Western blot analysis of indicated protein expression in BMSCs from mice with indicated genotypes (at 6-MO). GAPDH was used as a loading control. **d** Quantification analyses of the data in **c**, *$p < 0.05$, ***$p < 0.001$. mean ± SD *n* = 3. Mann–Whitney *U* test.

mice were subjected to RT-PCR analyses with *P16^Ink4a* and *P53* transcripts––both markers of senescence. Interestingly, both *P16^Ink4a* and *P53* were increased in the cortex and TA muscles, but not kidney or liver, of 6-MO *TgAPP_swe^OCN* mice (Supplementary Fig. 10a–e). These results suggest brain-region and tissue selective senescence-like phenotypes in *TgAPP_swe^OCN* mice.

**Diminished behavior phenotypes and brain pathology in *TgAPP_swe^OCN* mice treated with senescence inhibitor.** To determine if the increased senescence and SASPs in *TgAPP_swe^OCN* mice contribute to the brain and behavior deficits, we treated *TgAPP_swe^OCN* mice with Dasatinib (D) + Quercetin (Q), or Veh control (10% PEG 400), because the

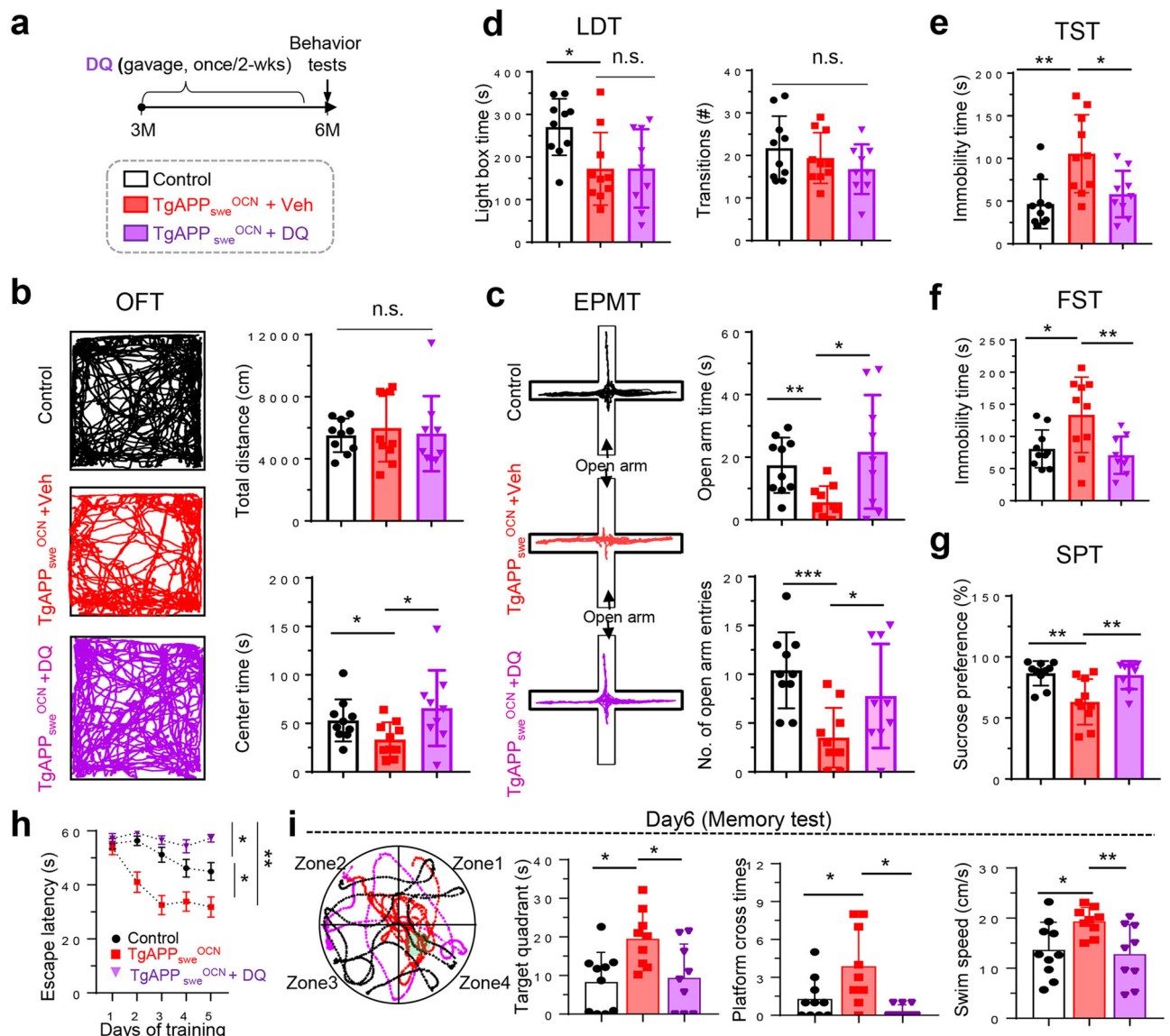

**Fig. 7 Diminished behavior phenotypes in *TgAPP_swe^OCN* mice treated with senescence inhibitors. a** Schematic diagram of experimental design. 6-MO control (*LSL-APP_swe*, *n* = 10 males) and *TgAPP_swe^OCN* mice were treated with Veh (10%PEG 400) (*n* = 10 males) or DQ (D 5 mg/kg, Q 50 mg/kg, dissolved in 10% PEG 400, once per two weeks) (*n* = 9 males), starting at age of 3-MO, and then subjected to indicated behavior tests at 6-MO. **b** OFT: Representative tracing images and quantifications of the total distance and the center duration time were shown. n.s. not significant, *$p < 0.05$. **c** EPMT: Representative tracing images and quantifications of the open arm duration time and entries were shown. *$p < 0.05$, **$p < 0.01$, ***$p < 0.001$. **d** LDT: Quantifications of the time spent in the light room and the number of transitions into the light room were shown. n.s. not significant, *$p < 0.05$. **e** TST, **f** FST, and **g** SPT were shown. *$p < 0.05$, **$p < 0.01$. **h–i** MWM: the latency to reach the hidden platform during the training period (**h**), and representative tracing image and quantification of the time spent in the target quadrant, platform crossing time and swim speed (**i**) were shown. *$p < 0.05$, **$p < 0.01$. One-way ANOVA followed by Tukey post hoc test. All data were presented as mean ± SD.

combination of D + Q is a well examined effective senolytic drug in animal studies[56,57]. We first treated cultured OB progenitors (BMSCs) from *TgAPP_swe^OCN* mice with D + Q. As expected, the senescence markers (SA-β-gal, P53, and P16^Ink4a) and the SASP-like factors (e.g., *Il1b, Il6, Cxcl1, Ccl5,* and *Tgfb1*) were all decreased in D + Q treated APP_swe^+ OB-progenitors (Supplementary Fig. 11), verifying D + Q's inhibitory effects on OB-senescence. We then administered D + Q to *TgAPP_swe^OCN* mice as illustrated in Fig. 7a. Remarkably, nearly all the behavior phenotypes, including depression (by TST, FST, and SPT), anxiety (by OFT and EPMT), and improved spatial learning and memory (by MWM) in *TgAPP_swe^OCN* mice (at 6-MO) were all diminished by D + Q treatments (Fig. 7b–i), providing evidence for senescence as a

potential pathological mechanism for these behavior changes. Notice that D + Q treatments had little effect on the anxiety-like behavior assessed by LDT (Fig. 7d), implicating additional mechanism(s) underlying this event.

Moreover, the GFAP^+ reactive astrocytes, IBA1^+ cells, and SASP-like factors (e.g., *Il1b, Tnfa*, but not *Il10* or *Mmp3*) in *TgAPP_swe^OCN* cortex were attenuated (Supplementary Fig. 12a–c), and the impaired hippocampal DG neurogenesis in *TgAPP_swe^OCN* mice was restored (Supplementary Fig. 12d, e) by D + Q treatments. In aggregates, these results suggest that APP_swe-induced senescence and SASPs are likely to prompt cortical brain inflammation and glial activation, which may underlie the behavioral phenotypes in 6-MO *TgAPP_swe^OCN* mice.

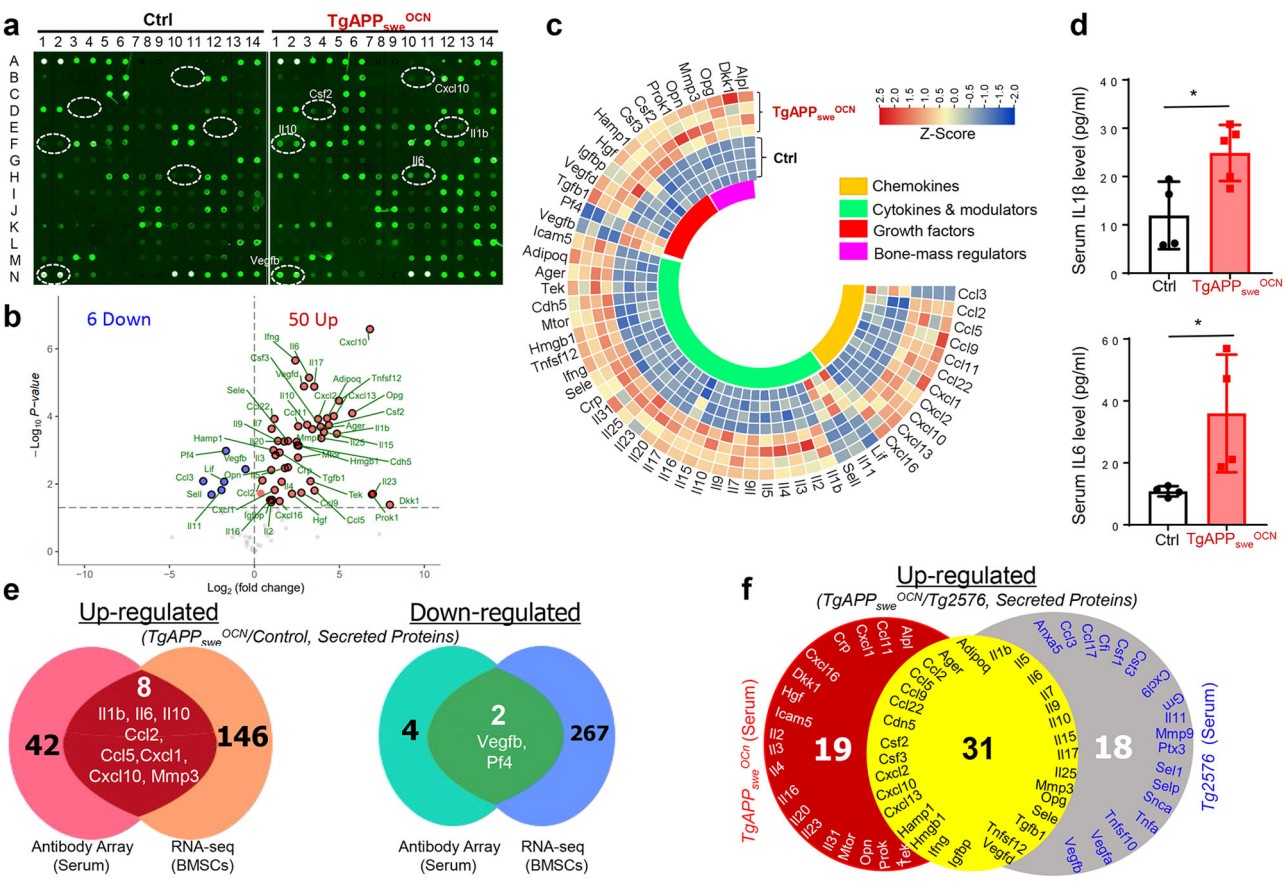

**Fig. 8 Increased cytokines and chemokines in *TgAPP_swe^OCN* serum samples. a** Representative images of serum L-Series label-multiplex antibody arrays of ~7-MO control and *TgAPP_swe^OCN* mice. **b** Volcano plots analysis of **a**. **c** Heat map of data in **a**. $n = 4$, significant difference was set at $p < 0.05$. **d** Elisa assays of serum IL1β and IL6 levels in ~7-MO control and *TgAPP_swe^OCN* mice. The data were presented as mean ± SD ($n = 4$ mice). *$p < 0.05$ by Student's $t$ test. **e** Comparison between this antibody array with secreted factors by RNA-seq of purified Tdtomato⁺ BMSCs. **f** Comparison of the changes (upregulated secreted proteins in Tg2576 over control mice) to those detected in *TgAPP_swe^OCN* mice.

**Systemic inflammation in *TgAPP_swe^OCN* mice likely due to APP_swe-induced OB-senescence and SASPs**. To further understand how APP_swe-induced OB-senescence and SASPs contribute to the brain pathology and behavior changes in *TgAPP_swe^OCN* mice, we speculate that APP_swe induced OB-senescence and SASPs contribute to systemic inflammation, which promotes brain inflammation and behavior changes. To test this speculation, we addressed the following questions.

First, are the increased SASPs (such as cytokines and chemokines) in APP_swe⁺ OB progenitors released and traveled through the circulation system of *TgAPP_swe^OCN* mice to induce the systemic inflammation? Using multiplexed antibody-based arrays to screen for altered serum/plasma proteins in *TgAPP_swe^OCN* mice (~7-MO) compared to their litter-mate control mice (*LSL-APP_swe*), increases in chemokines (CCL2, 5, 9, 11, 22, CXCL1, 2, 10, 13, 16), cytokines (IL1β, 2, 3, 4, 5, 6, 7, 9, 10, 15, 16, 17, 20, 23, 25, 31), and cytokine modulators (HMGB1, MTOR, CDH5) in the serum samples of *TgAPP_swe^OCN* mice (Fig. 8a–c), indicating a systemic inflammation. The increases in serum IL1β and IL6 levels in *TgAPP_swe^OCN* mice were verified by ELISA analyses (Fig. 8d). Notice that 8 up-regulated proteins (IL1β, 6, 10; CCL2, 5; CXCL1, 10 and MMP3) and 2 down-regulated proteins (VEGFB and PF4) were identified not only by the serum antibody array assay, but also by the RNA-seq analysis of OB progenitors (Fig. 8e), suggesting that many of the serum cytokines come from the APP_swe-induced OB-derived SASPs.

Second, is APP_swe in OCN-Cre⁺ cells a key contributor to the systemic inflammation? Although APP_swe is largely expressed in

OB-lineage cells of *TgAPP_swe^OCN* mice (Fig. 1), we cannot rule out the potential contribution of APP_swe's weak expression in the hippocampal dDG to systemic inflammation. To this end, we examined the serum inflammatory cytokines and chemokines in mice (*LSL-APP_swe*) injected with AAV-CaMKII-Cre or AAV-GFP into their dDGs; and the Cre-injected mice exhibited similar levels of APP_swe/Aβ42 in the hippocampus compared to 12-MO *TgAPP_swe^Ocn* mice (Supplementary Fig. 8g). Using a small-scale antibody array containing antibodies against multiple SASP-like pro-inflammatory cytokines and chemokines (Supplementary Fig. 13a), little to no change was detected between the serum samples from the Cre and GFP injected mice (Supplementary Fig. 13a, b). These results thus eliminate the possibility of dDG APP_swe/Aβ42 contribution to the systematic inflammation, supporting APP_swe in OCN-Cre⁺ OB-lineage cells as a major contributor of systemic inflammation. We also measured serum inflammatory factors in Tg2576 mice, a well-studied AD animal model that expresses *APP_swe* ubiquitously[24], using multiplexed antibody-based arrays, and compared the changes (upregulated secreted proteins in Tg2576 over control mice) with *TgAPP_swe^OCN* mice. Among 49 upregulated secreted proteins in Tg2576 mice, 31 (~63%) were increased in *TgAPP_swe^OCN* mice (Fig. 8f), providing additional support for the view.

Third, is the systemic inflammation results from the APP_swe induced-senescence and SASPs? Measuring serum SASP-like cytokines and chemokines in *TgAPP_swe^OCN* mice treated with and without D + Q, as illustrated in Fig. 7a, demonstrate that many cytokines (IL1β, 2, 23, 27) and chemokines (CCL2, 11 and

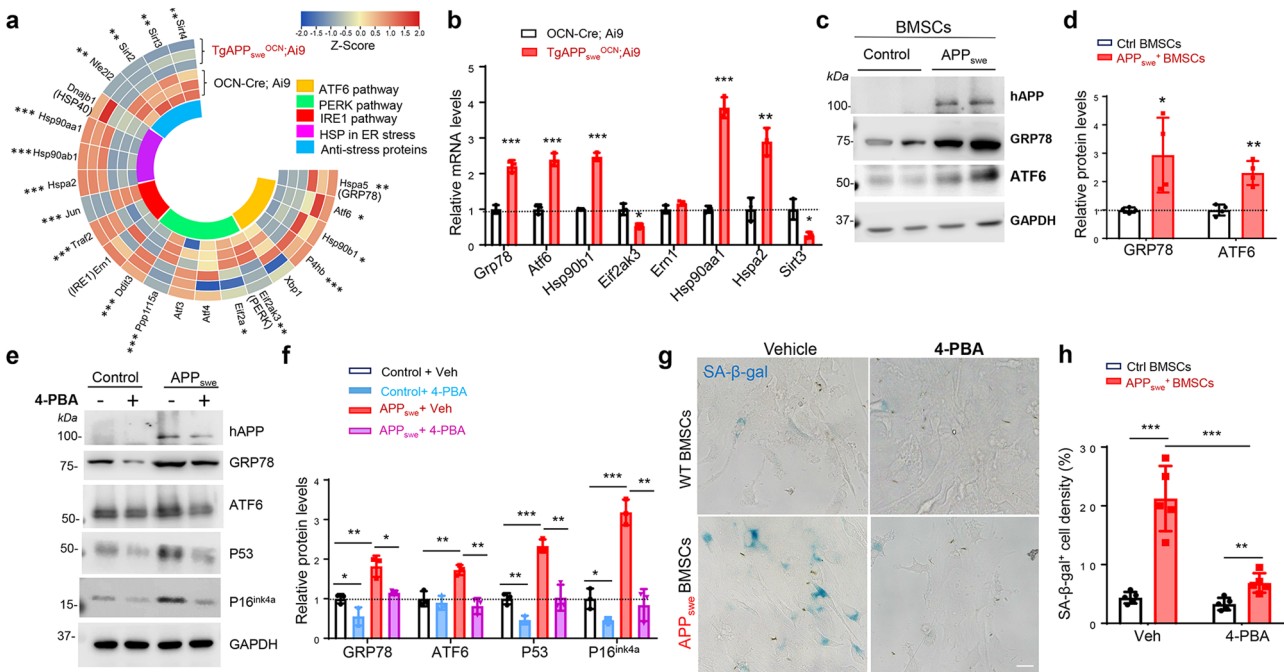

**Fig. 9 APPswe induction of OB-senescence via ER stress. a** Heat map of differentially expressed ER stress or anti-stress related genes identified by RNA-seq in control (OCN-Cre; Ai9) and TgAPPswe[OCN]; Ai9 Td[+] OB-progenitors (detail analysis was described in Methods). **b** RT-PCR analysis of ER stress-related genes Grp78, Atf6, Hsp90b1, Eif2ak3, Ern1, Hsp90aa1, and Hspa2 and anti-stress related gene Sirt3 gene expression in purified Td[+] BMSCs from 6-MO control (OCN-Cre; Ai9) and TgAPPswe[OCN]; Ai9 mice, *p < 0.05, **p < 0.01, ***p < 0.001, mean ± SD, n = 3, Mann–Whitney U test. **c** Western blot analysis of indicated protein expression in BMSCs from mice with indicated genotypes (at 6-MO). GAPDH was used as a loading control. **d** Quantification of data in **c**, *p < 0.05, **p < 0.01. mean ± SD, n = 4, Student's t test. **e** Western blot analysis of indicated protein expression in BMSCs from 6-MO control and TgAPPswe[OCN] with or without 0.25 mM 4-PBA (4-Phenylbutyric acid) treatment. **f** Quantification analyses of the data in **e**, *p < 0.05, n = 3. **g** SA-β-gal staining of 6-MO control and TgAPPswe[OCN] BMSCs with vehicle (Veh)(PBS) and 4-PBA treatment, respectively, scale bar, 20 μm. **h** Quantification of SA-β-gal[+] cell densities in **g** (mean ± SD; n = 5, **p < 0.01, ***p < 0.001). Two-way analysis of variance test was used in **f** and **h**.

CXCL1, 2) were increased in serum samples of TgAPPswe[OCN] mice treated with Veh, but decreased in the mice with D + Q treatments (Supplementary Fig. 13c, d). Together, these results suggest that the systemic inflammation in TgAPPswe[OCN] mice is likely in large due to the APPswe-induced OB-senescence and SASPs.

**Induction of ER stress-driven OB-senescence by expression of APPswe, but not APPwt or APPlon.** To understand how APPswe in OB-lineage cells induces senescence, we re-analyzed the RNA-seq data (APPswe[+] vs control OB progenitors) and found that, in addition to the increases in mRNAs of senescence genes, the transcripts of ER stress genes (e.g., Grp78, Atf6, and Hsp90) were elevated in APPswe[+] OB-progenitor cells (Fig. 9a, b). The increase in ER stress proteins (e.g, GRP78 and ATF6) were further verified by Western blot (Fig. 9c, d). To investigate the relationship between APPswe-induced ER stress and senescence, we treated APPswe[+] OB-progenitors with 4-PBA (4-Phenylbutyric acid), an inhibitor of ER stress[58]. 4-PBA treatments abolished the increases of the senescence marker proteins P16[Ink4a], P53, and SA-β-gal (Fig. 9e–h), suggesting that APPswe likely increases OB-senescence by inducing ER stress.

Notice that ER stress can be induced by the overexpression of membranous proteins[59]. It thus is necessary to determine if the increased ER stress in APPswe[+] cells results from its over expression. To this end, MC3T3 cells (an OB cell line) expressing APPwt-YFP (wild type), APPswe-YFP, and APPlon-YFP were examined. MC3T3 cells expressing APPswe-YFP, but not APPwt-YFP or APPlon-YFP, showed an obvious increase in GRP78 (an ER stress sensor) (Supplementary Fig. 14a, b), indicating a more dramatic effect on ER stress by APPswe-YFP and demonstrating its specificity.

Additionally, a more prominent co-localization of GRP78 with APPswe-YFP than those with APPwt-YFP or APPlon-YFP was observed (Supplementary Fig. 14a, c). Moreover, APPswe-YFP had an increased co-localization with EEA1, an early endosome marker, but decreased co-location with GM130, a marker for Trans-Golgi, compared with those of APPwt-YFP or APPlon-YFP (Supplementary Fig. 14d–g). These results demonstrate APPswe's distinctive cellular features in its increase of GRP78 and its subcellular localizations. Finally, the senescence marker, SA-β-gal, was selectively increased in MC3T3 cells expressing APPswe-YFP, but not APPwt nor APPlon (Supplementary Fig. 14h–i), providing additional support for the specificity of the detrimental effects by APPswe, but not by the overexpression of APPwt or APPlon.

## Discussion

Patients with AD often have osteopenia or osteoporosis[3–10]. The lower bone mineral density is often reported in the earliest clinical stages of AD patients (both men and women) and associated with their brain atrophy and memory decline[8]. However, it remains unclear if the AD patients carrying the Swedish mutations have osteoporosis-like deficit. Here, using TgAPPswe[OCN] mouse model that selectively expresses APPswe largely in the OB-lineage cells, we found that APPswe in OB-lineage cells induces senescence and SASPs, which appear to be a key contributor of systemic inflammation, and thus promote anxiety- and depression-like behaviors in TgAPPswe[OCN] mice. Our studies also suggest that the senescence may be insufficient to induce the cognitive decline detected in 12-MO TgAPPswe[OCN] mice, which may be associated with a weak expression of APPswe/Aβ42 in the dDG neurons of the hippocampus. These observations, summarized in Fig. 10a, lead to a working hypothesis depicted in

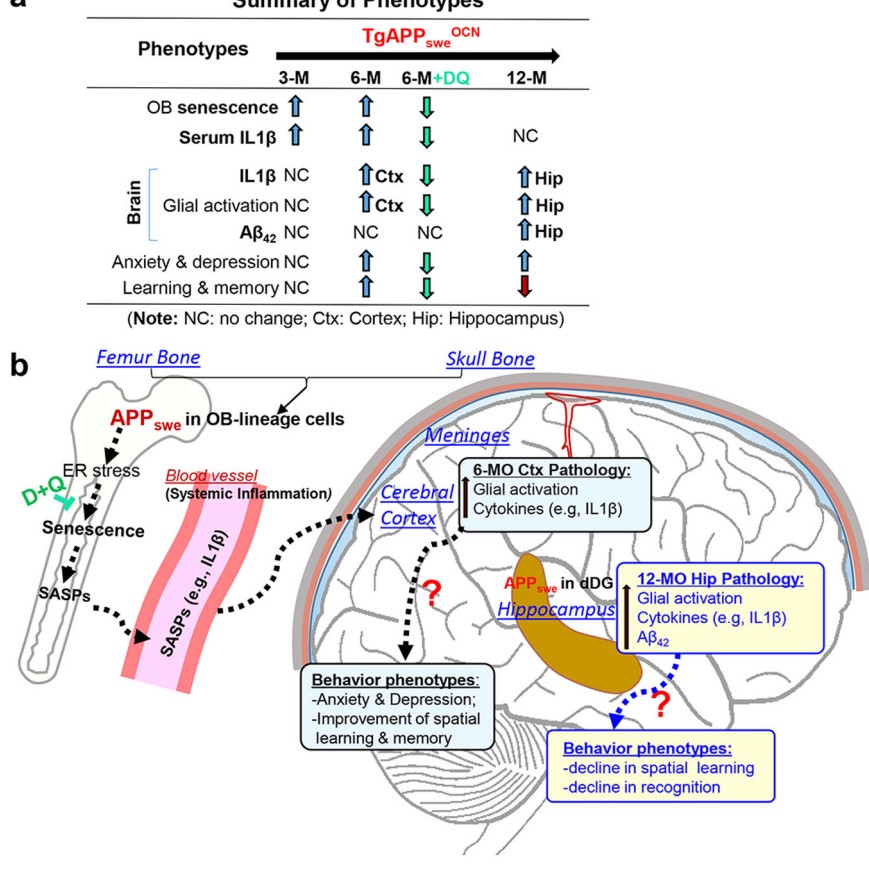

**Fig. 10 Summary and working hypothesis for APP$_{swe}$ in OB-lineage cells to regulate brain-pathology and behavior changes. a** Summary of phenotypes detected in *TgAPP$_{swe}$OCN* mice at indicated ages. **b** Illustration of the working model.

Fig. 10b, and opens a box of additional questions for future investigations.

A prerequisite to a better understanding of the mechanisms of *TgAPP$_{swe}$OCN* mice' brain/behavior phenotypes is to reveal where exactly the APP$_{swe}$ is expressed. *TgAPP$_{swe}$OCN* mice are generated by crossing *OCN-Cre* with the *LSL-hAPP$_{swe}$* mice, and thus the *APP$_{swe}$* expression is controlled not only by the CAG promoter in *LSL-hAPP$_{swe}$* mice (for its mRNA expression), and but also by the *OCN-Cre* dependent removal of LSL (for *hAPP$_{swe}$* protein expression)[23]. Although *OCN-Cre* mice express Cre largely in OB-lineage cells[60], our recent study demonstrates the Cre activity in neurons at the dDG, olfactory bulb, and cerebellum of the brain in *OCN-Cre* mice[29]. Our further studies in this paper lead us to conclude that *hAPP$_{swe}$* or *OCN-Cre* is largely expressed in the OB-lineage cells, but weakly expressed in the dDG neurons, in 12-MO *TgAPP$_{swe}$OCN* mice (Fig. 1b–g). We thus believe that the cortical brain and behavior phenotypes in 6-MO *TgAPP$_{swe}$OCN* mice are likely induced by the APP$_{swe}$ in OB-lineage cells. However, it is possible that the weak APP$_{swe}$/Aβ$_{42}$ expression in aged (12-MO) dDG hippocampal neurons contributes to the inflammation phenotypes in the mutant hippocampus and the cognitive decline (Fig. 10b).

How does APP$_{swe}$ in OB-lineage cells induce brain pathology? Several lines of evidence support the hypothesis that APP$_{swe}$-induced OB-senescence and SASPs may underlie its effects on the brain, particularly the cortex, via systemic inflammation (Fig. 10b). Many SASP-like proteins were induced in cultured APP$_{swe}$+ OB progenitors and increased in serum samples of *TgAPP$_{swe}$OCN* mice (Figs. 5 and 8). Cultured APP$_{swe}$+ OB progenitors and MC3T3 cells showed increased senescence cells (Fig. 6 and Supplementary Fig. 14h–i)[52,61]. While the OB-

senescence phenotypes were temporally associated with APP$_{swe}$-induced bone-deficits[23], they occurred earlier than brain deficits, in *TgAPP$_{swe}$OCN* mice (Fig. 6). The inhibition of senescence in *TgAPP$_{swe}$OCN* mice diminished nearly all the brain and behavior phenotypes (Fig. 7 and Supplementary Fig. 12). In line with this hypothesis are the multiple literature reports that demonstrate cellular senescence as tightly linked to skeleton and brain aging and various degenerative diseases, including AD[62–67], and the use of senolytic drugs to attenuate the disease process has been shown in several animal models of AD[68,69].

In terms of the systemic inflammation, while it can be induced by deficits in multiple organs, our results suggest that APP$_{swe}$-induced senescence and SASPs in OB-lineage cells appear to be a key contributor to this event. Many (31 over 49, ~63%) upregulated SASP-like factors detected in serum samples of *TgAPP$_{swe}$OCN* mice were also detected in Tg2576 mice (Fig. 8f). Although APP$_{swe}$ is weakly expressed in the dDG neurons of *TgAPP$_{swe}$OCN* mice (Fig. 1), examining the serum inflammatory cytokines and chemokines in mice (*LSL-APP$_{swe}$*) injected with AAV-CaMKII-Cre or AAV-GFP into their dDGs showed an increase in APP$_{swe}$/Aβ42 in the hippocampus of Cre injected mice (Supplementary Fig. 8g), but a comparable level of serum cytokines and chemokines between Cre and GFP injected mice (Supplementary Fig. 13a, b). Treatments with senescence inhibitors (D + Q) abolished nearly all the increased inflammatory cytokines in the serum samples of *TgAPP$_{swe}$Ocn* mice (Supplementary Fig. 13c, d). These results thus eliminate the possible contribution of the APP$_{swe}$/Aβ$_{42}$ at the dDG to systematic inflammation, and support the view.

How does APP$_{swe}$ in OB-lineage cells induce senescence and SASPs? We believe that APP$_{swe}$-induced ER stress may underlie this process for the following reasons. First, expressing APP$_{wt}$,

APP$_{swe}$, or APP$_{lon}$ in osteoblastic cell line, MC3T3 cells, results in an increased of β-gal$^+$ SnCs specifically in APP$_{swe}$$^+$, but not APP$_{wt}$$^+$ or APP$_{lon}$$^+$, cells (Supplementary Fig. 14h–i), although APP or Aβ levels were increased in all three types of cells. These results not only suggest the specificity of APP$_{swe}$ in the induction of the senescence, but also implicate Aβ's insufficiency or independency to this event. Second, APP$_{swe}$, compared to APP$_{wt}$ or APP$_{lon}$, exhibited distinctive features in its subcellular localizations and its induction of ER-stress, in addition to senescence (Supplementary Fig. 14a–g), revealing an association between the selective induction of the ER stress and senescence by APP$_{swe}$, but not APP$_{wt}$ or APP$_{lon}$, in line with the view that APP$_{swe}$ is processed by β-secretase or BACE1 in Golgi-derived vesicles, and APP$_{wt}$ is cleaved in the endosomes[70]. Third, both RNA-seq and Western blot analyses showed that APP$_{swe}$$^+$ OB progenitors have increased expressions of not only senescence associated genes, but also ER stress genes (e.g., *Grp78*, *Atf6*, and *Hsp90*) (Fig. 9a–d); and treatment of APP$_{swe}$$^+$ OB progenitors with an ER stress inhibitor 4-PBA abolished the increase of senescence marker proteins P16$^{Ink4a}$, P53, and β-gal$^+$ SnCs (Fig. 9e–h), supporting the view for ER stress as a driver of senescence. Notice that Hashimoto et al. report an absence of ER stress responses in *App*$^{NL-G-F}$ (App knock-in mice harboring Swedish mutation) brain[59]. We thus speculate that this event may be cell type/tissue specific, and OB-lineage cells may be more sensitive to APP$_{swe}$ than neurons in its induction of ER stress.

Are senescence and SASPs induced by osteoblastic APP$_{swe}$ involved in the behavior changes observed in TgAPP$_{swe}$$^{OCN}$ mice? Our results suggest that they are likely contributors to anxiety and depression, but insufficient to cause cognitive decline. In addition to the temporal association between the increased SASPs and the behavior changes, inhibition of senescence and SASPs by D + Q diminishes nearly all the behavior changes in TgAPP$_{swe}$$^{OCN}$ mice at 6-MO (Fig. 7). Among the SASPs induced by APP$_{swe}$, IL-1β is noteworthy, because IL-1β is found to mediate bi-functions in regulating spatial learning and memory[71–73]. Expressing IL-1β in the brain (in particular, the cortex) exhibits enhanced spatial learning and memory in young adult, but not aged, mice[74], a similar behavioral phenotype examined in the TgAPP$_{swe}$$^{OCN}$ mice (Figs. 2g and 4 and Supplementary Fig. 5g). This IL-1β's function is also in agreement with numerous reports, that IL-1β is upregulated by long term potentiation (LTP) (an event critical for learning and memory)[75–77]. The overexpression of IL-1ra, an endogenous IL-1R antagonist or IL-1R KO (knock-out), blocks spatial memory[78,79] as well as LTP[73,80]. In the light of these reports, we speculate that the osteoblastic APP$_{swe}$, via increasing IL-1β, a key SASP, may improve hippocampal/cortex-dependent spatial learning and memory function in an age-dependent manner. We are also aware of controversial reports, which claim that IL-1β plays a detrimental role in regulating learning and memory[81,82]. While IL-1β plays a role in modulating learning and memory, its precise function appears to strongly depend on the site of IL-1β injection/increase, timing, and dosage[73,79]. Notice that *Il1b* was increased in the hippocampus but not the cortex of 12-MO TgAPP$_{swe}$$^{OCN}$ mice (Supplementary Fig. 5g–h); and such IL-1β increase was accompanied by elevated Aβ$_{42}$ and glial activation in the hippocampus, and cognitive decline behaviors (Supplementary Figs. 1b and 5 and Fig. 4g–i). We thus speculate that the hippocampal inflammation phenotype may be induced by the weak APP$_{swe}$/Aβ$_{42}$ expression in dDG hippocampal neurons, which may also impair cognitive function in 12-MO TgAPP$_{swe}$$^{OCN}$ mice (Fig. 10b). It would be of interest to further test this view in future experiments.

Finally, it is highly possible that complex mechanisms underlie APP$_{swe}$ regulation of brain and behavior phenotypes in TgAPP$_{swe}$$^{OCN}$ mice. In addition to IL-1β and TNFα, other SASPs and growth factors may also contribute to the brain pathology. In addition to senescence and SASPs, the weak expression of APP$_{swe}$/Aβ$_{42}$ in the OCN-Cre$^+$ dDG neurons may be exacerbated by systemic inflammation and be responsible for the hippocampal pathology and cognitive decline in aged (e.g., 12-MO) TgAPP$_{swe}$$^{OCN}$ mice. It is also noteworthy that while chronic inflammation is believed to be one of the environmental risk factors for AD development[30,83], our studies suggest that the chronic systemic inflammation associated with AD patients (either EOAD or LOAD) may be induced by a combination of AD genetic risk gene(s), a primary hit, and environmental risk factors (e.g., aging, infection), a secondary hit, in line with the two-hit hypothesis[84]. Further investigations that address how chronic inflammation is induced, how it promotes the brain pathology and behavior changes, and what is the function/contribution of APP$_{swe}$/Aβ$_{42}$ in dDG neurons to the AD development may gain more insights into the two-hit hypothesis and AD pathogenesis.

## Methods

**Mice.** The *LSL-APP*$_{swe}$ mice were generated using the pCCALL2 plasmid as described previously[23]. In brief, the transcription of *hAPP*$_{swe}$ in *LSL-APP*$_{swe}$ mice is controlled by the CAG promoter, but its translation is blocked by a loxP-stop-loxP sequence[23]. Thus, the expression of hAPP$_{swe}$ is controlled by both the CAG promoter and the Cre-dependent removal of LSL. The *OCN–Cre* mice were kindly provided by Tom Clemens (Johns Hopkins Medical School). *OCN-Cre; Ai9* and *TgAPP*$_{swe}$$^{OCN}$; *Ai9* mice were generated by crossing *Ai9* mice (from the Jackson Laboratory, donated by Dr. Hongkui Zeng, Allen Institute for Brain Science) with *OCn-Cre* and *TgAPP*$_{swe}$$^{OCN}$ mice, respectively. *Ai9* mice have a loxP-flanked STOP cassette preventing the translation of a CAG promoter-driven red fluorescent protein variant (tdTomato). Thus, tdTomato is expressed following Cre-mediated recombination. The *Tg2576* mice were purchased from Taconic, Hudson, NY, USA, which express human *APP695* with Swedish double mutations at KM670/671NL (*APP*$_{swe}$) under the control of a hamster prion promoter[24]. *5xFAD* transgenic mice were obtained from The Jackson Laboratory (MMRRC stock #34 840-JAX)[25] which express human *APP* and *PSEN*1 transgenes with five AD-linked mutations (the Swedish [K670N/M671L], Florida [I716V], and London [V717I] mutation in *APP*, and the M146L and L286V mutation in *PSEN1*) under the control of mouse *Thy*1 promoter. All mouse lines were backcrossed into *C57BL/6* background and housed in a room with a 12 h light/dark cycle and ad libitum access to water and rodent chow diet (Harlan Tekled S-2335). Control littermates were used in parallel for each experiment. All experimental procedures were approved by the Institutional Animal Care and Use Committee at Case Western Reserve University (IACUC, 2017–0121), according to the United States National Institutes of Health guidelines.

**Antibodies and chemicals.** The following primary antibodies were used and purchased as indicated below: Anti-hAPP (6E10, 803001, mouse) and anti-6E10 (Alexa Fluor® 647 anti-β-Amyloid, 1-16 Antibody, cat#803021) from biolegend (San Diego, California, USA); Anti-Amyloid Fibrils OC antibody (AB2286, rabbit) from EMD Millipore (Temecula, California, USA); Anti-DCX (SC-8066, goat) from Santa Cruz Biotech (Santa Cruz, California, USA); Anti-Ctip2 (ab18465, Rat), anti-IBA1 (ab178846, rabbit and ab5076, goat), anti-P16$^{ink4a}$ (ab211542, rabbit), and anti-P53 (ab26, mouse) from Abcam (Cambridge, Massachusetts, USA); Anti-S100β (287004, Guinea pig) from Synaptic System (Göttingen, Germany); Anti-Olig2 (p21954, rabbit), anti-GRP78 (PA1-014A, rabbit) and anti-EEA1 (PA1-063A, rabbit) from Invitrogen (Carlsbad, California, USA); Anti-ATF6 (NBP1-40256, mouse) from Novus biologicals (Centennial, CO, USA); anti-GM130 (610822, mouse) from BD biosciences (San Jose, CA, USA) and Anti-P21 (2947S, rabbit), anti-NEUN (12943S, rabbit), anti-GFAP (12389S, rabbit), and anti-GAPDH (97166S, mouse) from cell signaling (Danvers, Massachusetts, USA). Secondary antibodies were purchased from Jackson ImmunoResearch Laboratories (West Grove, Pennsylvania, USA). Dasatinib was from LC Laboratories (Woburn, MA, USA). Quercetin, polyethylene glycol 400, 4-PBA, DMSO, DAPI, and d 5-ethynyl-2'-deoxyuridine (EdU, a modified thymidine analogue that is incorporated into the DNA of dividing cells) were from Sigma Aldrich (St. Louis, MO, USA). All chemicals and reagents used in this study were of analytical grade.

**Immunofluorescence staining and image analysis.** Immunostaining was performed as described previously[29]. In brief, mice were anesthetized with isoflurane and were transcardially perfused with PBS (50 mL) followed by 4% (w/v) paraformaldehyde (PFA) in phosphate buffer (PBS) (pH 7.4) (50 ml) to remove intravascular plasma proteins. The dissected brains were post-fixed in 4% PFA at 4 °C overnight. Coronal sections (40 μm) were washed 3 times with PBS (10 min each) and treated with blocking reagent (10% Donkey Serum + 0.5% Triton 100×) for

1 h, then incubated overnight at 4 °C with the primary antibody. Sections were washed 3 times and incubated with corresponding conjugated secondary antibody for 1 h. DAPI was used for nucleus counter staining. Stained sections were imaged by confocal microscope at room temperature. Fluorescent quantification was performed using ZEN software according to the manufacturer's instructions (Carl Zeiss).

**Western blotting**. Western blotting was performed as described previously[85]. Brain tissues and cultured BMSCs were homogenized in modified RIPA buffer (50 mM Tris-HCl, pH 7.5, 150 mM NaCl, 1 mM EDTA,) containing 0.5% sodium deoxycholate, 0.1% SDS, 1 mM PMSF, 1 mM Na3VO4, 1 mM NaF, 1 mM DTT, and protease inhibitor cocktail (Millipore, 539134). Lysates were centrifuged at 10,000 x $g$ for 10 min at 4 °C to remove debris and to obtain homogenates. Samples were resolved by SDS-PAGE and transferred to a nitrocellulose membrane (1620112, Bio-Rad Laboratories). After incubation with 5% milk in TBST (10 mM Tris, 150 mM NaCl, and 0.5% Tween 20, pH 8.0) for 1 h, membranes were immunoblotted with indicated antibodies overnight at 4 °C. Membranes were washed with TBST three times and incubated with a 1:2000 dilution of horseradish peroxidase–conjugated anti–mouse or anti–rabbit antibodies for 1 h. Blots were washed with TBST three times and immunoreactive bands were visualized using the LI-COR Odyssey infrared imaging system. Intensity of immunoreactive bands were quantitated by using ImageJ (NIH).

**EdU injection and labeling**. Control (*LSL-APPswe*) and *TgAPP*$_{swe}$*$^{OCN}$* mice were given four intraperitoneal injections of EdU (50 mg/kg/time, 1 time/4 h) within 12 h. 12 hours after their last injection, mice were euthanized and transcardially perfused first with 50 ml of cold PBS and then with 50 ml of 4% PFA. The dissected brains were post-fixed in 4% PFA at 4 °C overnight. Coronal sections (40 μm) were obtained for staining. Cultured BMSCs were incubated with 10 μM EdU for 2 hours, and then cells were fixed with 4%PFA for 10 min. EdU staining was performed using a Clik-iT EdU imaging kit with Alexa-Fluor 488 (Invitrogen) following the manufacturer's instructions.

**Behavioral tests**. Mice (male) at ages of 3-, 6- or 12-MO (month old) were subjected to behavioral studies. Behavioral tests were done blind to genotypes or treatments. For all behavioral experiments, mice were transferred to the testing room 4 h before any test to acclimate to the environment. All behavioral instruments were cleaned with 70% ethanol prior to each trial.

Open field test (OFT), Elevated plus maze test (EPMT), and Light/dark transition test (LDT) were performed as described previously[29]. In brief, for OFT, each mouse was placed in a chamber ($L \times W \times H = 50 \times 50 \times 20$ cm) and its movement was monitored for 10 min using an overhead camera. Light intensity was about 150 lux. The video was analyzed by a tracking software (Etho Vision, Noldus). The total distance and center ($25 \times 25$ cm) duration time were quantified. For elevated plus maze test (EPMT), the EPM was placed 50 cm above the ground. Each mouse was initially placed in the center square facing one of the open arms ($L \times W = 60 \times 5$ cm). Light intensity was about 100 lux. Mice movement was recorded for 5 min using an overhead camera and tracking software (Etho Vision, Noldus). The time spent in the open arms and the number of open arm entries were quantified. For light/dark transition test (LDT), mouse was firstly placed in the dark compartment, overhead camera was turned on, and the door between lit and dark chambers was opened. Light intensity was about 200 lux in the lit chamber. 10 min of movement was recorded using a tracking software ((Etho Vision, Noldus). The time spent in the lit chamber and the number of transitions were quantified.

The tail suspension test (TST), forced swimming test (FST), and sucrose preference test (SPT) were performed as described previously[86]. For the TST and FST, the last 4-min of a 6-min test were analyzed, and the immobility time was measured directly. The sucrose preference test was carried out using a two-bottle choice procedure. Single housed mice were habituated to drink 2% (wt/vol) sucrose solution (dissolved in water) for 3 days, then mice were given access to the two pre-weighed bottles, one containing water and the other containing 2% sucrose solution. Bottle positions were changed every day and water and sucrose solution consumption was assessed daily for 4 days. The consuming ratio of sucrose over total solution consumed was used for measuring the sucrose preference.

The Morris water maze (MWM) was performed as previously described[87]. Specifically, a 120 cm pool and 10 cm platform were used for water maze and nontoxic bright white gel (Soft Gel Paste Food Color, AmeriColor) was added to the water to make the surface opaque and to hide the escape platform (1 cm below the surface). Mice were trained for 5 days, four trials per day with 20 min interval between trials and 60 s per trial to locate the hidden platform. Eight spatial cues were placed on the pool wall, visible for mice to find the hidden platform. On the 6th day, the platform was removed, and mice were placed into the pool at a new starting position. The time spent in each platform quadrant and the number of platform-crossing within 60 s were analyzed. The swim speed and the amount of time spent in each quadrant were quantified using the video tracking system (Noldus). The investigators were blind to genotype during data acquisition and analysis.

The Novel Object Recognition Task (NOR) was based on a previous published procedure[88]. It consists of a *habituation phase* followed by a *testing phase*. During the *habituation phase*, each mouse was allowed to freely explore the empty arena over two days. On the third day, the testing phase begun. Habituation consisted of one ten-minute session administered one per day. The *testing phase* consisted of a (1) *familiarization trial* followed by a (2) *test trial*. During the *familiarization trial*, a single mouse was placed in the arena containing two identical objects and released against the center of the opposite wall with its back to the objects. This was done to prevent coercion to explore the objects. Object interaction is defined as entrance into the object-containing zone resulting in direct or nearly direct object contact with the nose or whiskers. The test trials were administered after delays of 1-hour post-familiarization. The *test trial* was administered in the aforementioned way except that one sample object from the familiarization trial and one novel object were presented. During the test trials, time spent with novel object per total time with both objects as the novel object preference was quantified.

**AAV virus injection**. AAV9-CamkII-GFP (105541-AAV9) and AAV9-CamkII-Cre (105551-AAV9) were purchased from Addgene. Virus injection was performed as described previously[29]. In brief, male *LSL-APPswe* mice (4-month-old) were anesthetized with Ketamine/Xylazine (HENRY SCHEIN #056344), and the head was fixed in a stereotaxic device (David Kopf Instruments). After the antiseptic treatment, the skull was exposed and cleaned using 1% $H_2O_2$. Holes were drilled into the skull and viruses were bilaterally injected into DG at the coordinates relative to bregma: caudal: −2.06 mm; lateral:±1.3 mm; ventral: −1.75 mm. After injection, the needle was left in place for 5 min to allow for diffusion of injected viruses before being slowly withdrawn. For the following 5 days, mice were daily injected with Meloxicam to reduce pain. Injection locations were validated in each mouse after experiments.

**In vitro primary OB-progenitor (BMSCs) cultures**. OB-progenitor (BMSCs) culture was carried out following a standard protocol as described previously[28,85]. In brief, the whole bone marrow cells flushed out from long bones of mice with DMEM were filtered through a 70-mm filter mesh, washed, re-suspended, and then plated in 100-mm dishes with growth medium (DMEM plus 10% FBS), which were incubated at 37°C with 5% $CO_2$. 3 days later, the non-adherent cells were removed. The attached bone marrow cells were cultured with the growth medium for 7 days. These cells were passaged and cultured for another 3–6 days with the same growth medium. These cells, so called BMSCs, were used for Western blot, RT-PCR, and SA-β-gal staining.

**Flow cytometry analysis**. Flow cytometry analysis was done as previously described[28]. BMSCs were flushed from femurs and tibias of 6-MO *OCN-Cre; Ai9* and *TgAPPswe*$^{OCN}$*; Ai9* mice, the attached bone marrow cells were cultured with the growth medium for 7 days. These cells were passaged and cultured for another 3 days with the same growth medium. Then cell media were removed from culture dishes and cells were rinsed with PBS. Trypsin solution was added to incubate at 37 °C for 2 min. The detached adherent cells were centrifuged, and the pellet cells were washed with 1 ml cold PBS, and finally resuspended in 0.5 ml PBS with 1% FBS for flow cytometry analysis. Flow cytometric analysis was performed by use of a flow cytometer in CWRU core facility. Acquisition and analysis were performed by using FACSDiva 8.0.1 software (BD).

**Generation of plasmids of APP$_{WT}$-YFP, APP$_{swe}$-YFP, and APP$_{lon}$-YFP**. YFP-APPswe mutation (K670N/M671L, AAG ATG - AAC TTG) and YFP-APP-London point mutation (V717I, GTC - ATC) from the YFP-APP$_{WT}$ construct by using the Q5 Site-Directed Mutagenesis Kit (E0554S, New England Biolabs, Inc). The primers 'CTGAAGTGAACTTGGATGCAGAATTCCGACATG' and 'AGAT CTCCTCCGTCTTGATATTTG' were used to generate the K670N/M671L mutation, and the primers 'CATCACCTTGGTGATGCTGAAG' and 'ATGATCA CTGTCGCTATGACAAC' were used to generate the V717I point mutation.

**MC3T3 cell culture and transfections of YFP, APP$_{WT}$-YFP, APP$_{swe}$-YFP, and APP$_{lon}$-YFP plasmids**. MC3T3-E1 cells were grown in DMEM containing 10% (vol/vol) FBS, and 50 units/ml penicillin and streptomycin. Cells plated at $1 \times 10^4$/well onto 12-wells coverslips the day before transfection. Cells were transfected with control-YFP Vector, APP$_{WT}$-YFP, APP$_{swe}$-YFP, and APP$_{lon}$-YFP by Lipofectamine 3000 (Invitrogen). Forty-eight hours after transfection, cells were subjected to SA-β-gal staining and immunofluorescence staining.

**SA-β -gal staining**. Cultured OB progenitors and MC3T3 cells SA-β-gal staining was performed as previously reported[85]. SA-β-gal staining was performed using a SA-β-gal staining kit (Cell Signaling, #9860) according to the manufacturer's instructions.

**Elisa assay for IL1β, IL6, human Aβ$_{40}$, and human Aβ$_{42}$**. Blood samples were collected, allowed to clot for 30 min, and centrifuged for 10 min at 3000 rpm. Serum was frozen and aliquot at −80 °C until use. Serum IL1β was measured with Mouse IL-1 beta ELISA Kit (KE10003, Proteintech), following the manufacturers'

instruction. Serum IL-6 was measured with mouse IL-6 ELISA kit (550950, BD Biosciences), following the manufacturers' instruction. Serum, Brain and BMSCs homogenization was obtained for human $A\beta_{40/42}$ Elisa assay. Brain tissues were homogenized as previously described[87]. Human $A\beta40$ and $A\beta42$ level in serum, brain (300 μg in total protein) and BMSCs (50 μg in total protein) homogenates were measured by the $A\beta_{40}$ human ELISA kit (Invitrogen, catalog #KHB3481) and the $A\beta_{42}$ human ELISA kit (Millipore, catalog #EZHS42), respectively. Their concentrations were determined by comparing readings against their standard curves.

**L-Series label-multiplex antibody arrays**. Mice blood samples were collected and allowed to clot for 30 min at room temperature and centrifuged for 15 min at 3000 rpm. Serum was frozen and aliquot at −80 °C until use. The antibody arrays were performed using an L-Series Glass Slide antibody arrays kit (AAM-SERV-LG, Raybiotech, USA) according to the manufacturer's instructions. In brief, the serum was dialyzed before the biotin-labeling step. The primary amine of the proteins in the sample was biotinylated, followed by dialysis to remove free biotin. The newly biotinylated sample was added onto the glass slide and incubated at room temperature. After incubation with Fluorescent Dye-Strepavidin, the signals were visualized by fluorescence.

**Mouse cytokine array**. Serum samples were collected as described above. Cytokines were measured with Mouse Cytokine Array Panel A (ARY006, R&D Systems). In Brief, the serum was mixed with a cocktail of biotinylated detection antibodies. The sample/antibody mixture was then incubated with the Mouse Cytokine Array membrane. Any cytokine/detection antibody complex present was bound by its cognate immobilized capture antibody on the membrane. Following a wash to remove unbound material, streptavidin–horseradish peroxidase and chemiluminescent detection reagents were added sequentially. Light was produced at each spot in proportion to the amount of cytokine bound.

**RNA isolation and qPCR**. Total RNA was isolated from brain tissues and BMSCs by using the RNeasy Mini Kit (QIAGEN, Cat No. 74104), and purified RNA (1–5 μg) was used for cDNA synthesis with the Revert Aid First Strand cDNA Synthesis Kit (Thermo Scientific, # K1621). cDNA products were subjected for subsequent quantitative PCR (qPCR) using the QuantiFast SYBR Green PCR Kit (204057; QIAGEN) with a qPCR System (StepOne Plus). Primers used were as follows: hAPP, 5′-GCCCTTCTCGTTCCTGAC-3′ and 5′-TCGCAAACATCCA TCCTC-3′; OCN-Cre, 5′-CAAATAGCCCTGGCAGATTC-3′ and 5′-TGATACA AGGGACATCTTCC-3′; mAPP, 5′-TCCGTGTGATCTACGAGCGCAT-3′ and 5′-GCCAAGACATCGTCGGAGTAGT-3′; Vps35, 5′-GACTTCGCTGATGAAC AGAGCC-3′ and 5′-CAGTGTGAAGCGAATCCGCTGA-3′; Trem2, 5′-CTACC AGTGTCAGAGTCTCCGA-3′ and 5′-CCTCGAAACTCGATGACTCCTC-3′; Apoe, 5′-GAACCGCTTCTGGGATTACCTG-3′ and 5′-GCCTTTACTTCCGTC ATAGTGTC-3′; Sorl1, 5′-GAACACCTGTCTCCGAAACCAG-3′ and 5′-CGG AACTGAGTGTCTGCATCAC-3′; Ptk2b, 5′-CTGGAGAGCATCAACTGTGTG C-3′ and 5′-GATGGGTAGACGTGTCACAGAG-3′; Msx2, 5′-AAGACGGAGC ACCGTGGATACA-3′ and 5′-CGGTTGGTCTTGTGTTTCCTCAG-3′; Runx2, 5′- CCTGAACTCTGCACCAAGTCCT-3′ and 5′- TCATCTGGCTCAGATAG GAGGG-3′; Sp7 (Osterix), 5′- GGCTTTTCTGCGGCAAGAGGTT-3′ and 5′- CGC TGATGTTTGCTCAAGTGGTC-3′; Nfatc1, 5′- GGTGCCTTTTGCGAGCAGT ATC-3′ and 5′- CGTATGGACCAGAATGTGACGG-3′; Col1a1, 5′- CCTCAGGG TATTGCTGGACAAC-3′ and 5′- CAGAAGGACCTTGTTTGCCAGG-3′; Spp1 (Osteopontin), 5′- GCTTGGCTTATGGACTGAGGTC-3′ and 5′- CCTTAGAC TCACCGCTCTTCATG-3′; Bglap (Osteocalcin), 5′- GCAATAAGGTAGTGAAC AGACTCC-3′ and 5′- CCATAGATGCGTTTGTAGGCGG-3′; Sparc (Osteo-nectin), 5′- CACCTGGACTACATCGGACCAT-3′ and 5′- CTGCTTCTCAGT-GAGGAGGTTG-3′; Bmp2, 5′- TGTGAGGATTAGCAGGTCTTTGC-3′ and 5′- C TCGTTTGTGGAGCGGATGT-3′; Lrp4, 5′- GTGTGGCAGAACCTTGACAGT C-3′ and 5′- TACGGTCTGAGCCATCCATTCC-3′; Ctnnb1(beta-catenin), 5′- GT TCGCCTTCATTATGGACTGCC-3′ and 5′- ATAGCACCCTGTTCCCGCAAA G-3′; Tnfsf11 (Rankl), 5′- ATCCCATCGGGTTCCCATAA-3′ and 5′-TCCGTT GCTTAACGTCATGTTAG-3′; Tnfrsf11b (Opg), 5′-GGCCTGATGTATGCCCTC AA-3′ and 5′-GTGCAGGAACCTCATGGTCTTC-3′; Alpl, 5′- CCAGAAAGACA CCTTGACTGTGG-3′ and 5′- TCTTGTCCGTGTCGCTCACCAT-3′; Il1b, 5′- T GGACCTTCCAGGATGAGGACA-3′ and 5′- GTTCATCTCGGAGCCTGTAG TG-3′; Il6, 5′- CTTGGGACTGATGCTGGTGACA-3′ and 5′- TGGTAGGTGGTATC CTCTGTGA-3′; Il10, 5′- CGGGAAGACAATAACTGCACCC-3′ and 5′- CGGT TAGCAGTATGTTGTCCAGC-3′; Tnfa, 5′- GGCGGTGCCTATGTCTCA-3′ and 5′- CCTCCACTTGGTGGTTTGT-3′; Cxcl1, 5′- TCCAGAGCTTGAAGGTGTTG CC-3′ and 5′- AACCAAGGGAGCTTCAGGGTCA-3′; Ccl5, 5′- ACCACTCCCT GCTGCTTT-3′ and 5′- ACACTTGGCGGTTCCTTC-3′; Tgfb1, 5′-ACCGCAACA ACGCCATCT-3′ and 5′-GGGCACTGCTTCCCGAAT-3′; Csf2, 5′- AACCTCCT GGATGACATGCCTG-3′ and 5′- AAATTGCCCCGTAGACCCTGCT-3′; Mmp3, 5′- CTCTGGAACCTGAGACATCACC-3′ and 5′- AGGAGTCCTGAGAGATT TGCGC-3′; Grp78, 5′-TGTCTTCTCAGCATCAAGCAAGG-3′ and 5′-CCAACAC TTCCTGGACAGGCTT-3′; Atf6, 5′-GTCCAAAGCGAAGAGCTGTCTG-3′ and 5′-AGAGATGCCTCCTCTGATTGGC-3′; Hsp90b1, 5′-GTTTCCCGTGAGAC TCTTCAGC-3′ and 5′-ATTCGTGCCGAACTCCTTCCAG-3′; Eif2ak3, 5′-CCG

ATGTCAGTGACAACAGCTG-3′ and 5′-AAGACAACGCCAAAGCCACCAC-3′; Ern1, 5′-GGCTACCATTATCCTGAGCACC-3′ and 5′-CTCCTTCTGGAACTGT TGGTGC-3′; Hsp90aa1, 5′-GCTTTCAGAGCTGTTGCGGTAC-3′ and 5′-AAAGG CGGAGTTAGCAACCTGG-3′; Hspa2, 5′-GCACCTTCGATGTGTCCATCCT-3′ and 5′-TGGCTGACCATACGGTTGTCGA-3′; Sirt3, 5′-GCTACATGCACGGT GTGTCGAA-3′ and 5′-CAATGTCGGGTTTCACAACGCC-3′; GAPDH, 5′-AA GGTCATCCCAGAGCTGAA-3′ and 5′-CTGCTTCACCACCTTCTTGA-3′. Each sample was repeated at least 3 times, and the mRNA level was normalized to GAPDH using the 2-$\triangle\triangle$Ct method.

**Bulk RNA-sequencing**. Total RNAs were extracted from purified Td$^+$ OB-progenitor cells from OCN-Cre; Ai9 and $TgAPPswe^{OCN}$; Ai9 mice by flow cytometer. RNA Integrity Number (RIN) was accessed for every sample, and the samples were considered qualified with RIN > 2. These RNA samples were then subjected to RNA-seq analyses by BGI America (Cambridge, MA) using the DNBseq platform. Firstly, we removed the reads mapped to rRNA and obtained the raw data with 52.47 Mb reads. After filtering low-quality, adaptor-polluted and high content of unknown base reads in the sequencing reads, 51.9 Mb clean reads were obtained per sample on average. Then clean reads were mapped to reference genome using HISAT2. On average 92.91% reads were mapped and the uniformity of the mapping result for each sample suggests that the samples were comparable. Comparisons to RNAseq were normalized to fpkm values. DEseq2 was used and the PossionDis algorithms detected the differential expression genes (DEGs). The Benjamini and Hochberg (BH) correction was applied to adjust p-value. DEGs were determined with adj.P.value ≤ 0.05 and |Log2 fold change| ≥ 1. Normalized RNA-seq data were provided in Supplementary Data 1. Heatmap was generated by TBtools software. Gene expression profiles were Z-transferred. Secreted protein database was obtained from http://proteomics.ysu.edu/secretomes/animal/index.php. Gene Ontology (GO) enrichment analysis was performed by GO database (http://www.geneontology.org/). GO terms with p-value ≤ 0.05 were defined as significantly enriched.

**Statistics and reproducibility**. All data were expressed as means ± SD. For in vivo studies, three to ten male mice per genotype per assay were used. For in vitro cell biological and biochemical studies, each experiment was repeated at least three times. Statistical analyses were performed using GraphPad Prism 7.0. Mann–Whitney $U$ test or unpaired Student's $t$ test was used to compare data from two groups. For multiple comparisons of three or more groups of samples, ANOVA was used. The significance level was set at $P < 0.05$ (*$P < 0.05$, **$P < 0.01$, ***$P < 0.001$).

**Reporting summary**. Further information on research design is available in the Nature Research Reporting Summary linked to this article.

## Data availability
Source data of figures are provided in Supplementary Data 1. Sequencing data that support the findings in this study have been assigned Gene Expression Omnibus accession number GSE186827. The uncropped Western blots are provided in Supplementary Figs. 15–22. The data that support the findings of this study are available from the corresponding author upon reasonable request.

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

## Acknowledgements
We thank members in MeiXiong laboratories for helpful discussions, suggestions, and technical help. This study is supported in part by the National Institutes of Health (AG045781, AG051773, and AG051510 to W.-C.X.) and the Department of Veterans Administration (to W.-C.X). We also thank Meisel Family and InMotion at Cleveland, Ohio.

## Author contributions
J.P. and W.-C.X. designed the project and wrote the manuscript. J.P. and D.S. performed behavioral tests, virus injection, immunostaining, Western blot, and data analysis. D.L. performed MWM tests and data quantification. X.R. performed RNA-seq data analysis. L.X. generated the plasmids encoding APP$_{\text{lon}}$-YFP and APP$_{\text{swe}}$-YFP and did transfection of MC3T3 cells and data analysis; H.G. did L-Series label-multiplex antibody arrays. L.Y. helped with virus injection and data analysis. Y.L. performed mice genotyping and kept mouse lines. C.J. performed BMSCs culture. W.-C.X. helped data analysis, interpretation, and supervised the project.

## Competing interests
The authors declare no competing interests.
