## [Transparent Peer Review File · Communications Biology]

Reviewers' comments:

Reviewer #1 (Remarks to the Author):

The manuscript by Pan et al. aims to understand the link between dementia and osteoporotic deficits in mouse models. This is an unmet research area and was built nicely upon the previous work from the same group. They have previously shown that young adult Tg2576 mice, carrying Swedish mutant amyloid precursor protein (APP^{swe}) in these transgenic mice, show osteoporotic deficit with reduced bone formation and increased bone resorption. In the present study, Pan and colleagues focused on their studies using transgenic mice overexpressing APP^{swe} in osteoblast-lineage cells, called TgAPP^{swe}OCN mice, which expressed APP^{swe} in osteoblast-lineage cells but not in the cortex and hippocampus. Intriguingly, authors showed that these mice developed anxiety and depression-like behaviors as well as brain pathology such as increases in GFAP positive astrocytes, IBA1+ microglial cells, proinflammatory cytokines. These authors further explored mechanistic insight and showed that expressing APP^{swe} in osteoblast-lineage cells (OCN-Cre positive osteoblasts) resulted in elevated cellular senescence and senescence-associated secretory phenotypes (SASPs). Inhibition of cellular senescence by its inhibitors diminished nearly all the phenotypes in TgAPP^{swe}OCN mice. Overall, results were obtained through comprehensive sets of experiments in this study and supportive to their conclusion of AD as a systematic disorder. In my assessment, this study is very novel and important, considering the solid evidence that links osteoblastic APP^{swe} in contribution to the AD-associated brain phenotypes. Experimental designs in this study are rigorous, statistical analysis is adequate and presented data are striking. I have only a few minor concerns.

1. While senescence in osteoblasts induced by APP^{swe} expression is convincing, it is not clear how this occurs. Is it APP or A β or in what form? Authors should elaborate or comment on in the discussion.
2. It would be of interest to discuss whether Tg2576 mice express APP^{swe} in osteoblast-lineage cells.
3. Any amyloid plaque-like pathology in TgAPP^{swe}OCN BMSCs or brains.

Reviewer #2 (Remarks to the Author):

The manuscript by Pan et al., claims that expression of APP (harboring the Swedish mutation) in mice osteoblast (OB) lineage cells causes an Alzheimer's disease (AD)-like pathology evident by glial activation and behavioral deficits. The authors use a Cre-lox mouse model characterized by overexpression of APP^{swe} in OB cells. The mice exhibit a marked increased level of inflammatory mediators, senescence of OBSCs, glial activation and behavioral deficits. Their conclusion is that APP^{swe} in OB cells promotes senescence and systemic inflammation which consequently results in AD-like pathology evident at 6 months of age. Whereas the overall findings are solid, there are a few critical questions which should be addressed before the study suits the publication in *Comm. Biology*.

1. The animal model used:

- a. What's the physiological relevance of overexpressing the mutant APP^{swd} in OBs? The fact that AD patients suffer from osteoporosis or osteopenia does not by itself justify it as these issues are common in elderly individuals and most likely are more robust in AD patients given their enhanced aging process, systemic inflammation etc.
- b. Are the findings unique to APP^{swe} or they can be recapitulated with non-mutated human APP or other AD-related APP mutants?
- c. The authors show that although the tdTomato is expressed in various brain regions, the APP protein is rarely expressed. However, although much lower levels are expressed in the brain as compared with OBSCs, small amounts can be detected and may cause the observed mild glial activation and behavioral deficits.
- d. Since the model is tissue-specific but not conditional, how the authors can rule out developmental aberrations, especially considering that APP^{swe} is overexpressed.

2. Systemic inflammation: the authors show a marked elevation in serum levels of inflammatory mediators which by themselves can cause glial activation and behavioral deficits. This is not a feature of AD patients beside the slightly enhanced low-grade inflammation seen in elderly individuals. Furthermore, it is well known that systemic inflammation can cause cognitive deficits; in AD it was coined as the two-hit hypothesis. It is thus unclear why the authors propose that their findings underlie the pathophysiology of AD.

3. Senescence: The authors show that APP^{swe} overexpression in OBs induce senescence which presumably cause the pronounced systemic inflammation but there is no insight into the mechanism. In general, this aspect could be of interest provided that mutant APP is unique in causing such senescence when expressed at physiological levels. In addition, is senescence observed among glial cells in the brain as was recently evident in various models? If overexpressed in glial cells or neurons, will it cause a similar effect? Is senescence evident on other tissues such as the liver? The use of senolytic drugs to attenuate the disease process in animal models of AD has been shown in several animal models of AD.

4. Behavior: as aforementioned dysregulated inflammation can cause some behavioral abnormalities including anxiety. Is there any indication for memory loss and a decline in cognitive function?

Overall, the findings showing that senescence caused by APP^{swe} in OBs can lead to systemic inflammation and behavioral deficits are potentially interesting with additional mechanistic insights. In the current form of the manuscript, beside the use of one of the human mutant APP that cause early onset AD, the biological relevance of the model used to AD is yet unclear.

Reviewer #3 (Remarks to the Author):

In this work, authors provided experimental evidence that expression of APP^{swe} in osteoblast-lineage cells is related to psychiatric phenotype and increased neuroinflammation. Although their findings are new and interesting, several substantial concerns exist to be addressed to convince readers to accept their claims. I think this study is not suitable for publication in Communications Biology.

1. They utilized APP transgenic mouse models that overexpress APP ubiquitously or in osteoblast (OB)-lineage cells. Overexpression of APP protein can cause several artifacts that can influence the results and may lead to incorrect conclusions. They should confirm the results in the non-overexpression models such as App^{NL} knock-in mice that were developed by Takaomi C. Saido in RIKEN, Japan.

2. It is difficult to generalize their findings to most of AD patients because they don't have Swedish mutation, which accounts for a quite small proportion of AD patients. Supporting evidence from human patients is critically lacking.

3. The mechanism(s) that osteoblastic APP^{swe} worsen cognitive dysfunction is(are) not clear. Neuroinflammation in AD is complicated, and might be protective at a certain stage of the disease.

4. It's not clear how expression of APP^{swe} in OB-lineage cells cause cellular senescence and increase of particular cytokines and chemokines.

5. How about senescence-related phenotypes in other peripheral tissues?

Other major concerns

1. Only A β 40 is measured in the brain and serum. Aggregation-prone A β 42 should be also measured.
2. What is the reason that neuroinflammation is exacerbated only in the cortex, but not in hippocampus of TgAPPsweOCN mice?
3. They examined only in young mice (6-month-old is the oldest). How about these changes in aged TgAPPsweOCN mice?
4. What is the mechanism(s) that hippocampal DG neurogenesis is reduced in TgAPPsweOCN mice while neuroinflammation in hippocampus is not affected?
5. Expression level of APP and A β after injection of AAV-CamkII-Cre in LSL-APPswe mice should be assessed.

Response to Reviewer #1's comments:

We thank Reviewer 1 for his/her comments that *“The manuscript by Pan et al. aims to understand the link between dementia and osteoporotic deficits in mouse models. This is an unmet research area and was built nicely upon the previous work from the same group. They have previously shown that young adult Tg2576 mice, carrying Swedish mutant amyloid precursor protein (APP_{swe}) in these transgenic mice, show osteoporotic deficit with reduced bone formation and increased bone resorption. In the present study, Pan and colleagues focused on their studies using transgenic mice overexpressing APP_{swe} in osteoblast-lineage cells, called TgAPP_{swe}OCN mice, which expressed APP_{swe} in osteoblast-lineage cells but not in the cortex and hippocampus. Intriguingly, authors showed that these mice developed anxiety and depression like behaviors as well as brain pathology such as increases in GFAP positive astrocytes, IBA1+ microglial cells, proinflammatory cytokines. These authors further explored mechanistic insight and showed that expressing APP_{swe} in osteoblast-lineage cells (OCN-Cre positive osteoblasts) resulted in elevated cellular senescence and senescence associated secretory phenotypes (SASPs). Inhibition of cellular senescence by its inhibitors diminished nearly all the phenotypes in TgAPP_{swe}OCN mice. Overall, results were obtained through comprehensive sets of experiments in this study and supportive to their conclusion of AD as a systematic disorder. In my assessment, this study is very novel and important, considering the solid evidence that links osteoblastic APP_{swe} in contribution to the AD-associated brain phenotypes. Experimental designs in this study are rigorous, statistical analysis is adequate and presented data are striking. I have only a few minor concerns.”*

We also appreciate Reviewer 1 for his/her constructive comments below. Additional experiments have been performed to address nearly all the questions. A point-to-point response is below.

1. *“While senescence in osteoblasts induced by APP_{swe} expression is convincing, it is not clear how this occurs. Is it APP or A β or in what form? Authors should elaborate or comment on in the discussion.”*

Response: Good points! Additional experiments have been done to address these questions.

Is the senescence induced by APP, A β , or other form of APP? To address this question, we **first** examined senescence associated (SA)- β -gal⁺ cells in MC3T3 cells (an osteoblastic cell line) expressing human APP_{wt} (wild type), APP_{swe} (Swedish mutant), or APP_{Lon} (London mutant). Interestingly, SA- β -gal⁺ cells were detected only in APP_{swe}⁺, but not APP_{wt}⁺ nor APP_{Lon}⁺, cells (see revised supplementary Fig. 9), although APP and A β levels were increased in cells expressing APP_{wt}, APP_{swe}, or APP_{Lon}. These results demonstrate the specificity of APP_{swe} and implicate A β 's in-sufficiency or in-dependency in this event. We **further** tested the view of A β 's in-dependency by use of APP^{-/-} osteoblasts, in which, APP is knocked out, and thus A β is un-detectable. Remarkably, similar phenotypes, such as increased SA- β -gal⁺ cells and elevated expressions of senescence markers, p16^{Ink4a} and p53, as those of APP_{swe}⁺ osteoblasts were detected in APP^{-/-} osteoblast (OB)-lineage cells (see Fig. 1 below). These results suggest the necessity of APP in osteoblasts to prevent senescence, uncovering a physiological function of APP, and support the view for A β -independent induction of the cellular senescence. Note that young adult APP^{-/-} mice also show similar bone deficits (e.g., reduced bone formation and bone mass) as those of

TgAPP_{swe}^{OCN} mice^{1,2}, implicating that APP_{swe} induced senescence and bone phenotypes may result from its negative inhibition of endogenous APP. We will further test this view in future experiments.

How does osteoblastic APP_{swe} induce senescence? To address this question, we re-analyzed our RNA-seq data (APP_{swe}⁺ vs control OB-lineage cells) and found that in addition to the increases in mRNAs of senescence associated genes, the transcripts of ER stress genes (e.g., Grp78, Atf6, and Hsp90) were elevated in APP_{swe}⁺ OB-lineage cells (see revised Fig. 8a). We further verified the increased expression of Grp78 and Atf6, both ER stress proteins, in APP_{swe}⁺ osteoblasts by RT-PCR and Western blot analyses (see revised Fig. 8b-d). Notice that the Grp78 was also increased in APP^{-/-} OB-lineage cells (see Fig. 1c-d below). Moreover, we examined the potential relationship between APP_{swe}-induced ER stress and senescence by treatment of APP_{swe}⁺ OB-lineage cells with 4-PBA (4-phenylbutyric acid) (an inhibitor of ER stress). Inhibition of ER-stress by 4-PBA markedly reduced the levels of senescence marker proteins, p16^{Ink4A} and p53, and the SA-β-gal⁺ cells (see revised Fig. 8e-h), supporting the view for APP_{swe} to induce senescence by promoting ER stress. These new data are included in revised Fig. 8, and Supplementary Fig. 9, and described in Results (see pages 12-13) and Discussion (see page 16). However, the results obtained from APP^{-/-} OB-lineage cells are shown here for your reference (see Fig. 1 below), but not in revised manuscript, because the current manuscript contains 10 Figures, and 14 supplementary Figures (reached the maximum numbers of Figures that the Journal is allowed), and these data are not directly relevant to the current experimental objectives.

Fig.1. Increased cellular senescence in APP^{-/-} OB-lineage cells. (a) Senescence associate (SA)-β-gal staining of BMSCs (bone marrow stromal cells) derived from 6-MO control (WT) and APP^{-/-} mice. Scale bar, 50μm. (b) Quantification of data in (a), the SA-β-gal⁺ cell density is shown (mean ± SD; n=3). **p<0.01. (c) Western blot analysis using indicated antibodies in lysates of BMSCs from mice with indicated genotypes (at 6-MO). (d) Quantification of data in (c). The data were presented as mean ± SD, n=3. *p<0.05, **p<0.01, ***p < 0.001. Student's t test.

2. "It would be of interest to discuss whether Tg2576 mice express APP_{swe} in osteoblast lineage cells."

Response: Yes, APP_{swe} is highly expressed in osteoblast lineage cells derived from young adult Tg2576 mice. This result has been shown in our previous publications (see Cui et al, J Bone Miner Res. 2011; Xia et al, J Bone Miner Res. 2013), which is now described in revised Introduction (see page 4).

3. "Any amyloid plaque-like pathology in TgAPP_{swe}^{OCN} BMSCs or brains".

Response: Good point! We have addressed this question by ELISA analysis of A β _{40/42} and immunostaining analysis of both bone cells/bone and brain samples from TgAPP_{swe}^{OCN} mice (at ages of 6- and 12-MO). Whereas ELISA showed increases of A β ₄₀ and A β ₄₂ levels in BMSCs, A β ₄₂ was slightly increased in the hippocampus of TgAPP_{swe}^{OCN} mice at age of 12-MO, not 6-MO (see revised Fig. 1f-g and supplementary Fig. 1a-b), and little to no 6E10⁺ A β plaque was detected in 12-MO TgAPP_{swe}^{OCN} bone and brain sections (see revised supplementary Fig. 1c-d). As a control, many 6E10⁺ A β plaques were detected in brain sections of 5XFAD mice (4.5 MO) (see revised supplementary Fig. 1e). These results are described in revised manuscript (see pages 6-7).

Response to Reviewer #2's comments:

We thank Reviewer 2 for his/her comments that *"The manuscript by Pan et al., claims that expression of APP (harboring the Swedish mutation) in mice osteoblast (OB) lineage cells causes an Alzheimer's disease (AD)-like pathology evident by glial activation and behavioral deficits. The authors use a Cre-lox mouse model characterized by overexpression of APP^{swe} in OB cells. The mice exhibit marked increased levels of inflammatory mediators, senescence of OBSCs, glial activation and behavioral deficits. Their conclusion is that APP^{swe} in OB cells promotes senescence and systemic inflammation which consequently results in AD-like pathology evident at 6 months of age. Whereas the overall findings are solid, there're a few critical questions which should be addressed before the study suits the publication in Comm. Biology."*

We also appreciate Reviewer 2 for his/her constructive comments. A point-to-point response to his/her questions is below.

1. The animal model used:

a. *"What's the physiological relevance of overexpressing the mutant APP^{swe} in OBs? The fact that AD patients suffer from osteoporosis or osteopenia does not by itself justify it as these issues are common in elderly individuals and most likely are more robust in AD patients given their enhanced aging process, systemic inflammation etc."*

Response: Thank Reviewer 2 for raising this question, which gives us the opportunity to emphasize the physiological/pathological relevance of our studies.

First, as noted by the reviewer, patients with AD often suffer from osteoporosis or osteopenia. However, little are known regarding the underlying mechanisms of AD association with bone loss and their relationships. One view, as indicated by the reviewer, is that both AD and osteoporosis share common environmental risk factors, such as aging and systemic inflammation etc., and thus, both diseases could be linked. Another view is that osteoporosis or osteopenia could be a consequence of neurodegeneration in AD patients, as patients with AD often exercise less with little movement activity (a key activity for bone health). While these clinical and epidemiological observations support a degree of co-morbidity of both diseases, it remains poorly understood what their relationships are, and how they are linked together.

Second, a growing list of genetic risk genes has been recently identified in patients with late onset AD (LOAD). Intriguingly, many of the LOAD risk genes are expressed not in neurons, but in immune cells, and encode proteins that regulate immune response and bone-mass homeostasis. For examples, TREM2 (triggering receptors expressed on myeloid cells-2), an AD risk gene, is highly expressed in microglia/macrophages and osteoclasts (**OC**), but little in neurons; and it regulates not only microglial phagocytosis, but also OC-mediated bone remodeling³⁻⁵. ApoE, another genetic risk factor for AD, is also identified as a risk gene for osteoporosis⁶⁻⁸. These genetic studies suggest multiple common genetic denominator(s), in addition to environmental risk factors (aging and inflammation), underlying AD and osteopenia/osteoporosis development.

Third, we chose Swedish mutant APP (**APP_{swe}**) to address the question-how AD is linked with osteoporosis for the following reasons. **I)** **APP_{swe}** is one of the earliest mutants identified in patients with early-onset AD (EOAD). Although **APP_{swe}** is detected in small fractions of AD patients, its functions in A β production in the brain and in promoting AD pathogenesis have been well studied in multiple animal models. For examples, Tg2576 and 5XFAD, both well-characterized AD animal models, express **APP_{swe}** under the control of prion and Thy1 promoter, respectively^{10,11}. **II)** Much AD research has been focused on the impact of A β on the brain, even though **App** or **APP_{swe}** is known to be expressed not only in the brain, but also in periphery tissues, including osteoblast (**OB**)-lineage cells^{2,9}. Note that the **APP_{swe}** in Tg2576 mice is expressed ubiquitously, not only in the brain, but also in periphery tissues, including OB cells^{2,9}. While investigating phenotypes in **APP_{swe}** based animal models have provided valuable insights into A β pathology in the brain and impairments in mouse cognitive functions, the functions of **APP_{swe}** in periphery tissues, such as OBs, remain poorly understood. **III)** Our previous examinations of bone structures in Tg2576 mice have revealed early-onset osteoporotic deficits, months before any brain-pathologic defect that can be detected^{2,9}. Knocking out APP (in **APP^{-/-}** mice) or selective expression of **APP_{swe}** in osteocalcin (**OCN**) promoter driven Cre (**OCN-Cre**)⁺ osteoblasts [in **TgAPP_{swe}^{OCN}** mice] recapitulated the osteoporotic defect in Tg2576 mouse model^{1,2}. These observations argue against the view for the bone deficits as a consequence of neuro-degeneration, and raise an interesting question- could problems in the bone cells contribute to AD pathology in the brain? The current paper reports our results towards addressing this question. These points have been included in revised Introduction (see pages 3-4).

b. *“Are the findings unique to APP_{swe} or they can be recapitulated with non-mutated human APP or other AD-related APP mutants?”*

Response: Good point! As responded to Reviewer 1’s first question, we have examined senescence phenotype (SA- β -gal staining, and/or western blot analyses of p16^{ink4a} and p53, markers of senescence) in osteoblasts expressing **APP_{wt}**, **APP_{swe}**, **APP_{lon}** (London mutation), or OB-lineage cells derived from **APP^{-/-}** mice. Interestingly, the senescence phenotypes were detectable in **APP_{swe}⁺** and **APP^{-/-}** osteoblasts, but not in **APP_{wt}** or **APP_{lon}** expressing osteoblasts (see revised supplementary Fig. 9, and Fig. 1 in this response letter). These results suggest that the induction of the senescence in osteoblasts is **APP_{swe}** specific, and A β -independent, and uncover an APP’s physiological function. The similar phenotypes between **APP_{swe}** and **APP^{-/-}** osteoblasts (e.g., both showed increased ER-stress and senescence, decreased osteoblastogenesis, bone formation, and bone mass in mice^{1,2}) implicate that these phenotypes may be induced by **APP_{swe}**’s dominant negative inhibition of endogenous APP. We will further test this view in future experiments. These new data were included in revised Supplementary Fig.9 and described in Results (see page 12) and Discussion (see page 16).

c. *“The authors show that although the tdTomato is expressed in various brain regions, the APP protein is rarely expressed. However, although much lower levels are expressed in the brain as compared with OBSCs, small amounts can be detected and may cause the observed mild glial activation and behavioral deficits.”*

Response: Agree with the reviewer that it remains to be possible that the small amount of APP_{swe} in the brain may cause the observed mild glial activation and behavioral deficits. Although examinations of APP_{swe} and A β expression in 6-MO TgAPP_{swe}^{OCN} mice showed little protein, but a weak mRNA expression, in OCN-Cre⁺ brain regions (see revised Fig. 1), our recent studies in 12-MO TgAPP_{swe}^{OCN} mice demonstrate a slight increase of A β ₄₂, but not A β ₄₀, in their hippocampus, but not cortex (see revised Supplementary Fig. 1a-b). Interestingly, accompanied with the increased A β ₄₂ were hippocampal glial activation and elevated inflammatory cytokines, and cognitive dysfunctional behavior in 12-MO TgAPP_{swe}^{OCN} mice (see revised Supplementary Fig. 5 and Fig. 4g-i). These new results support the view for a weak APP_{swe} expression in DG hippocampal neurons that may contribute to the hippocampal glial activation/inflammation and behavior changes at age of 12-MO. These results have been described in revised Results (see pages 6-11) and Discussion (see page 17).

d. *“Since the model is tissue-specific but not conditional, how the authors can rule out developmental aberrations, especially considering that APP_{swe} is overexpressed.”*

Response: True, we couldn't rule out the developmental effect in phenotypes detected in TgAPP_{swe}^{OCN} mice. However, it is noteworthy that the OCN-Cre mice express Cre largely in mature/adult osteoblasts, and little in the embryonic stages^{12,13}. Nevertheless, we plan to address this issue by generating a new OCN-CreER mice using Crispr knock-in strategy, which is in the process.

2. Systemic inflammation:

“the authors show a marked elevation in serum levels of inflammatory mediators which by themselves can cause glial activation and behavioral deficits. This is not a feature of AD patients beside the slightly enhanced low-grade inflammation seen in elderly individuals. Furthermore, it is well known that systemic inflammation can cause cognitive deficits; in AD it was coined as the two-hit hypothesis. It is thus unclear why the authors propose that their findings underlie the pathophysiology of AD.”

Response: Yes, a marked elevation in serum levels of inflammatory mediators can cause glial activation and behavioral deficits, as pointed out by the Reviewer. Although this is not a typical pathological feature of AD patients, chronic inflammation is believed to be one of the environmental risk factors for AD development^{14,15}. Thus, it is important to address the questions-how chronic inflammation is induced in AD patients and how it contributes to AD pathology. Our studies lead us to speculate that the chronic inflammation associated with AD patients (either APP_{swe} induced EOAD or APP_{swe} independent LOAD) may be induced by a combination of AD genetic risk gene(s)(which act as a primary hit) and environmental risk factors (e.g., aging, infection,...)(as a secondary hit). Thus, understanding how chronic inflammation is induced and how it contributes to the brain pathology and behavior changes may open a window to view outside of the “brain box” of AD patients and gain more insights into AD pathogenesis. We have included these points in revised Discussion (see page 18).

3. Senescence:

“The authors show that APP_{swe} overexpression in OBs induce senescence which presumably cause the pronounced systemic inflammation but there is no insight into the mechanism. In

general, this aspect could be of interest provided that mutant APP is unique in causing such senescence when expressed at physiological levels.”

Response: As responded to Reviewer 1’s concerns (point 1), we have performed additional experiments to address this question. We have shown that the induction of senescence in osteoblasts was APP_{swe} specific because this phenotype was un-detectable in osteoblasts expressing APP_{wt} (wild type) or APP_{Lon} (London mutant) (see revised supplementary Fig. 9). To understand how APP_{swe} induces senescence in osteoblasts, we re-analyzed our RNA-seq data (APP_{swe}⁺ vs wild type control osteoblasts) and found that in addition to the increases in mRNAs of senescence associated genes, the transcripts of ER stress genes (e.g., Grp78, Atf6, and Hsp90) were elevated in APP_{swe}⁺ osteoblasts (see revised Fig. 8a). We then verified the increased Grp78 and Atf6 in APP_{swe}⁺ osteoblasts by both RT-PCR and Western blot analyses (see revised Fig. 8b-d). Moreover, we examined the potential relationship between APP_{swe}-induced ER stress and senescence by culturing APP_{swe}⁺ osteoblasts in the presence of 4-PBA (4-phenylbutyric acid) (an inhibitor of ER stress). Inhibition of the ER-stress by 4-PBA abolished the increased senescence marker proteins, p16^{Ink4A} and p53, and the SA-β-gal⁺ cells (see revised Fig. 8e-h), supporting the view for APP_{swe} to induce senescence by promoting ER stress. These new data were included in revised Fig. 8, and Supplementary Fig. 9, and described in Results (see pages 12-13) and Discussion (see page 16).

“In addition, is senescence observed among glial cells in the brain as was recently evident in various models?”

Response: Thank Reviewer 2 for raising this question. RT-PCR analysis showed increased expressions of p16^{Ink4a} and p53, markers of senescence, in the cortex, but not hippocampus, of 6-MO TgAPP_{swe}^{OCN} mice (see revised Supplementary Fig. 11a-b), suggesting a senescence phenotype in TgAPP_{swe}^{OCN} cortex. Further immunostaining analysis using antibody against p16^{Ink4a} failed to obtain specific signals. But, using antibody against p53 (a marker of senescence), we did detect a slight increase of P53⁺ cells in TgAPP_{swe}^{OCN} cortex. However, these P53⁺ cells were not GFAP⁺ astrocytes nor IBA1⁺ microglial cells (see revised Supplementary Fig. 11c-e). These new results are included in supplementary Fig. 11 and described in pages 12-13.

“If overexpressed in glial cells or neurons, will it cause a similar effect?”

Response: An interesting point! Yes, we did observe increased expressions of senescence markers, P16^{Ink4a} and P53, when APP_{swe} was expressed in neurons (by injection of AAV-CamkII-Cre) or when it was expressed in astrocytes (by injection of AAV-GFAP-Cre). Given the current manuscript containing 10 Figures, and 14 supplementary Figures, and considering this data not directly relevant to the current experimental objectives, these results are shown here for your review (see Fig. 2 below), but not in revised manuscript.

Fig. 2. Increased cellular senescence markers in *LSL-APP^{swe}* mice injected with AAV-CamkII-Cre or AAV-GFAP-Cre. (a) Schematic diagram of experimental design. AAV9-CamkII-GFP (control) and AAV9-CamkII-Cre were injected into the dorsal DG hippocampus of *LSL-APP^{swe}* mice (4-MO), respectively. (b) RT-PCR analysis of P16^{ink4a}, P53 gene expression in the hippocampus of indicated groups, **p<0.01, n=3. (c) Schematic diagram of experimental design. AAV5-GFAP-GFP (control) and AAV5-GFAP-Cre were injected into the cortex of *LSL-APP^{swe}*

mice, respectively. (d) RT-PCR analysis of P16^{ink4a}, P53 gene expression in the cortex of indicated groups. The data were presented as mean ± SD, n=3, *p<0.05, **p<0.01, by student's t test.

“Is senescence evident on other tissues such as the liver? The use of senolytic drugs to attenuate the disease process in animal models of AD has been shown in several animal models of AD.”

Response: Again, we thank Reviewer 2 for raising this question. We have examined senescence marker protein expressions in other tissues, including muscles, liver, and kidney, in addition to cortex and hippocampus. RT-PCR analysis showed increases in p16^{ink4a} and p53 not only in osteoblast lineage cells, but also in muscles and cortex from *TgAPP^{swe}^{OCN}* mice (at age of 6-MO) (see revised Supplementary Fig. 11a-f). But, both p16^{ink4a} and p53 were not increased in *TgAPP^{swe}^{OCN}* liver and kidney (see revised Supplementary Fig. 11g-h). These results thus suggest the tissue selectivity, implicating both muscle and cortex as vulnerable tissues/organs in *TgAPP^{swe}^{OCN}* mice (at 6-MO). These results are included in revised supplementary Fig.11 and described in page 12.

4. Behavior:

“as aforementioned dysregulated inflammation can cause some behavioral abnormalities including anxiety. Is there any indication for memory loss and a decline in cognitive function?”

Response: Good point! We have examined cognitive function of *TgAPP^{swe}^{OCN}* mice in various ages (3-, 6-, and 12-MO) by Morris water maze (MWM) (to access spatial learning and memory function) and novel object recognition (NOR) tests. Interestingly, age-dependent phenotypes were detected. No obvious difference in both MWM and NOR task performance was observed between *TgAPP^{swe}^{OCN}* and control mice at 3-MO (see revised Fig. 4a-c). However, at age of 6-MO, *TgAPP^{swe}^{OCN}* mice exhibited faster learning and better long-term memory during MWM tests, as compared with those of age matched control mice (revised Fig. 4d-e), suggesting an improvement in spatial learning and memory. But at age of 12-MO, impairments in both MWM and NOR tasks were detected in *TgAPP^{swe}^{OCN}* mice (revised Fig. 4g-i). These results are intriguing, demonstrating age-dependent changes in spatial and novel object learning and memory in *TgAPP^{swe}^{OCN}* mice. These new results are included in revised Fig. 4 and described in pages 10-11, 17.

Response to Reviewer #3's comments:

We thank Reviewer 3 for his/her comments that *"In this work, authors provided experimental evidence that expression of APP_{swe} in osteoblastlineage cells is related to psychiatric phenotype and increased neuroinflammation."* We also thank Reviewer 3 for, but respectively disagree with, below comments, *"Although their findings are new and interesting, several substantial concerns exist to be addressed to convince readers to accept their claims. I think this study is not suitable for publication in Communications Biology"*.

We believe that this study is of physiological/pathological significance, and thus suitable for publication in *Communications Biology*, for the following reasons.

First, as responded to Reviewer 2 (point 1), patients with AD often suffer from osteoporosis or osteopenia. Little are known regarding the underlying mechanisms of AD association with bone loss and their relationships. One view is that both AD and osteoporosis share common environmental risk factors, such as aging and systemic inflammation etc., and thus, both diseases could be linked. Another view is that osteoporosis or osteopenia could be a consequence of neurodegeneration in AD patients, as patients with AD often exercise less or have little movement activity (a key activity for bone health). While these clinical and epidemiological observations support a degree of co-morbidity of both diseases, it remains poorly understood what their relationships are, and how they are linked together.

Second, a growing list of genetic risk genes has been recently identified in patients with late onset AD (LOAD). Intriguingly, many of the LOAD risk genes are expressed not in neurons, but in immune cells, and encode proteins that regulate immune response and bone-mass homeostasis. For examples, TREM2 (triggering receptors expressed on myeloid cells-2), an AD risk gene, is highly expressed in microglia/macrophages and osteoclasts (**OC**), but little in neurons; and it regulates not only microglial phagocytosis, but also OC-mediated bone remodeling³⁻⁵. ApoE, another genetic risk factor for AD, is also identified as a risk gene for osteoporosis⁶⁻⁸. These genetic studies suggest multiple common genetic denominator(s), in addition to environmental risk factors (aging and inflammation), underlying AD and osteopenia/osteoporosis development.

Third, we chose Swedish mutant APP (**APP_{swe}**) to address the question-how AD is linked with osteoporosis for the following reasons. **I)** **APP_{swe}** is one of the earliest mutants identified in patients with early-onset AD (EOAD). Although APP_{swe} is detected in small fractions of AD patients, its functions in A β production in the brain and in promoting AD pathogenesis have been well studied in multiple animal models. For examples, Tg2576 and 5XFAD, both well-characterized AD animal models, express APP_{swe} under the control of prion and Thy1 promoter, respectively^{10,11}. **II)** Much AD research has been focused on the impact of A β on the brain, even though App or APP_{swe} is known to be expressed not only in the brain, but also in periphery tissues, including osteoblast (**OB**)-lineage cell Is^{2,9}. Note that the APP_{swe} in Tg2576 mice is expressed ubiquitously, not only in the brain, but also in periphery tissues, including OB cells^{2,9}. While

investigating phenotypes in APP_{swe} based animal models have provided valuable insights into A β pathology in the brain and impairments in mouse cognitive functions, the functions of APP_{swe} in periphery tissues, such as OBs, remain poorly understood. **III)** Our previous examinations of bone structures in Tg2576 mice have revealed early-onset osteoporotic deficits, months before any brain-pathologic defect that can be detected^{2,9}. Knocking out APP (in APP^{-/-} mice) or selective expression of APP_{swe} in osteocalcin (**OCN**) promoter driven Cre (**OCN-Cre**)⁺ osteoblasts [in TgAPP_{swe}^{OCN} mice] recapitulated the osteoporotic defect in Tg2576 mouse model^{1,2}. These observations argue against the view for the bone deficits as a consequence of neuro-degeneration, and raise an interesting question- could problems in the bone cells contribute to AD pathology in the brain? The current paper reports our results towards addressing this question. These points have been included in revised Introduction (see pages 3-4).

Finally, in reviewing the specific concerns raised by Reviewer 3 (below), we have performed additional experiments and are able to address nearly all of his/her concerns. A point-to-point response is below.

1. *“They utilized APP transgenic mouse models that overexpress APP ubiquitously or in osteoblast (OB)-lineage cells. Overexpression of APP protein can cause several artifacts that can influence the results and may lead to incorrect conclusions. They should confirm the results in the non-overexpression models such as AppNL knock-in mice that were developed by Takaomi C. Saido in RIKEN, Japan.”*

Response: Agree! It is important to verify our results in AppNL knock-in mice. However, this mouse line is not commercially available in US. We are in the process to obtain these mice from Dr. Saido (takaomi.saido@riken.jp, RIKEN Center for Brain Science, address: 2-1 Hirosawa, Wakoshi, Saitama 351-0198. JAPAN), who agrees to provide to us. Additionally, this mouse line is global knock-in, not floxed allele, and thus, it can't be useful to address tissue/cell type specific effects of the mutant APP.

On the other hand, we have made our efforts to address Reviewer's concern by examining senescence phenotype (SA- β -gal staining, and/or western blot analyses of p16^{Ink4a} and p53, markers of senescence) in osteoblasts expressing APP_{wt}, APP_{swe}, APP_{Lon} (London mutant), or osteoblasts derived from APP^{-/-} mice. Interestingly, the senescence phenotypes were detected in APP_{swe}⁺ and APP^{-/-} osteoblasts, but not in osteoblasts expressing APP_{wt} or APP_{Lon} (see revised supplementary Fig. 9 and Fig.1 in this response letter). These results suggest that the induction of the senescence in osteoblasts is APP_{swe} specific, and A β -independent, and uncover an APP's physiological function in preventing osteoblasts to undergo senescence. The similar phenotypes between APP_{swe} and APP^{-/-} osteoblasts (e.g., both showed increased senescence, decreased osteoblastogenesis and bone formation, and reduced bone mass in mice)^{1,2} implicate that these phenotypes may be induced by APP_{swe}'s dominant negative inhibition of endogenous APP. We hope to test this view and verify our results in AppNL knock-in mice when obtained these mice from Dr. Saido.

2. *“It is difficult to generalize their findings to most of AD patients because they don’t have Swedish mutation, which accounts for a quite small proportion of AD patients. Supporting evidence from human patients is critically lacking.”*

Response: Agree with Reviewer 3 that APP_{swe} accounts for a small proportion of AD patients and it is important to verify our results in human patients. On the other hand, it is noteworthy that APP_{swe} is often used for generation of AD animal models. For examples, both Tg2576 and 5XFAD mice express APP_{swe} under the control of prion and Thy1 promoter, respectively. Both APP and APP_{swe} in human and in Tg2576 mice are widely expressed, not only in the brain, but also in periphery tissues, including osteoblasts^{2,9}. Whereas investigating phenotypes in these APP_{swe} based animal models have provided valuable insights into A β brain pathology and impairments in mouse cognitive functions, it remains largely unclear APP’s physiological functions or APP_{swe}’s pathological roles in periphery organs, including bone cells. We hope that the publication of our paper, which uncovers an unrecognized link from bone to brain in mouse models, could stimulate physician scientists to test this view in AD patients. We hope to have this opportunity to collaborate with physician scientists to address this issue in future.

3. *“The mechanism(s) that osteoblastic APP_{swe} worsen cognitive dysfunction is(are) not clear. Neuroinflammation in AD is complicated and might be protective at a certain stage of the disease.”*

Response: Again, we agree with Reviewer 3 that “neuroinflammation in AD is complicated and might be protective at a certain stage of the disease”. Indeed, examining cognitive function of TgAPP_{swe}^{OCN} mice in various age groups (3-, 6-, and 12-MO) using Morris water maze (MWM)(to access spatial learning and memory function) and novel object recognition (NOR) tests showed age-dependent phenotypes. As responded to Reviewer 2 (Point 4), whereas no obvious difference in MWM and NOR task performance was detected in TgAPP_{swe}^{OCN} mice at 3-MO as compared with age- and gender-matched control mice (see revised Fig. 4a-c), at age of 6-MO, TgAPP_{swe}^{OCN} mice exhibited faster learning and better long-term memory, as compared with those of age- and gender-matched control mice (revised Fig. 4d-e), suggesting an improvement in spatial learning and memory. However, at age of 12-MO, impairments in both MWM and NOR tasks were detected in TgAPP_{swe}^{OCN} mice (revised Fig. 4g-i). These results are intriguing, demonstrating age-dependent changes in spatial and/or novel object learning and memory in TgAPP_{swe}^{OCN} mice, supporting the view that “neuroinflammation in AD is complicated, and might be protective at a certain stage of the disease”, and detrimental at other stage. These new results were included in revised Fig.4 and described in pages 10-11, 17.

4. *“It’s not clear how expression of APP_{swe} in OB-lineage cells cause cellular senescence and increase of particular cytokines and chemokines.”*

Response: Again, we have performed additional experiments to address this question. As responded to Reviewer 1 (Point 1) and Reviewer 2 (Point 3), we have provided evidence that the induction of senescence in osteoblasts was APP_{swe} specific, as this phenotype was un-detectable in osteoblasts expressing APP_{wt} (wild type) or APP_{Lon} (London mutant) (see revised supplementary Fig. 9). To further understand how APP_{swe} induces senescence in osteoblasts, we re-analyzed our RNA-seq data (APP_{swe}⁺ vs wild type control osteoblasts) and found that in addition to the

increases in mRNAs of senescence associated genes, the transcripts of ER stress genes (e.g., Grp78, ATF6, and Hsp90) were elevated in APP_{swe}⁺ osteoblasts (see revised Fig. 8a). We then verified the increased Grp78 (a sensor for ER stress) in APP_{swe}⁺ osteoblasts by Western blot analysis (see revised Fig. 8b-d). Moreover, we examined the potential relationship between APP_{swe}-induced ER stress and senescence by culturing APP_{swe}⁺ osteoblasts in the presence of 4-PBA (4-phenylbutyric acid) (an inhibitor of ER stress). Inhibition of the ER-stress by 4-PBA abolished the increases in senescence marker proteins, p16^{Ink4A} and p53 and SA-β-gal⁺ cells (see revised Fig. 8e-h), supporting the view for APP_{swe} to induce senescence by promoting ER stress. These new data were included in revised Fig. 8, and supplementary Fig. 9, and described in Results (see pages 12-13) and Discussion (see page 16).

5. “How about senescence-related phenotypes in other peripheral tissues?”

Response: Good question! A similar question is raised by Reviewer 2 (see Point 3). We have examined senescence marker protein expression in other tissues, including muscles, liver, kidney, cortex, and hippocampus. RT-PCR analysis showed increases of p16^{Ink4a} and p53 not only in osteoblast lineage cells, but also in muscles and cortex from TgAPP_{swe}^{OCN} mice (at age of 6-MO) (see revised Supplementary Fig. 11a-f). But, both p16^{Ink4a} and p53 were not increased in the TgAPP_{swe}^{OCN} liver and kidney (see revised supplementary Fig.11g-h), suggesting the tissue selectivity of this event, and implicating both muscles and cortex as vulnerable tissues/organs in TgAPP_{swe}^{OCN} mice. These results are included in revised supplementary Fig.11 and described in page 12.

Other major concerns

1. “Only Aβ40 is measured in the brain and serum. Aggregation-prone Aβ42 should be also measured.”

Response: Good suggestion! We have measured Aβ₄₂. Intriguingly, it was increased in the hippocampus (but not cortex or serum) of 12-, but not 6-MO, TgAPP_{swe}^{OCN} mice, supporting the view for a weak APP_{swe} expression in hippocampal DG neurons. This data is included in revised Fig. 1f-g, and Supplementary Fig. 1a-b, and described in pages 6-7.

2. “What is the reason that neuroinflammation is exacerbated only in the cortex, but not in hippocampus of TgAPP_{swe}^{OCN} mice?”

Response: Again, good question! Although it remains unclear regarding the reason(s) for neuroinflammation selectively in the cortex, but not hippocampus, of 6-MO TgAPP_{swe}^{OCN} mice, we speculate that APP_{swe}'s expression in skull bone cells (osteoblasts) in TgAPP_{swe}^{OCN} mice may be a key underlying reason, given skull bone cells' close association with the meningeal linings of the top layers of the cortex, where contain enriched blood and lymphatic vessels and glial cells, and considering the abundant OCN-Cre and APP_{swe} expression in skull bone cells in TgAPP_{swe}^{OCN} mice (see Fig. 3. Below). Note that these data are shown below for your reference, but not in revised manuscript, because the current manuscript contains 10 Figures, and 14 supplementary Figures, and this data is beyond the scope of the current experimental objectives.

Fig. 3. OCN-Cre and APP^{sw} expression in skull bone cells of *TgAPP^{sw}^{OCN}* mice. (a) Representative image of tdTomato in a skull bone coronal section from OCN-Cre; Ai9 mice (3-MO). Scale bar, 100 μ m. (b) Western blot analysis of hAPP protein in skull bone cells from mice with indicated genotypes (at 6-MO). (c) Quantification of the data in (b), ***p<0.001. (d) RT-PCR analysis of hAPP gene expression in control and *TgAPP^{sw}^{OCN}* skull bone cells. ***p<0.001. All the data were presented as mean \pm SD, n=3, and Student's t test.

Additionally, it is noteworthy that this brain-regional inflammation phenotype is age-dependent. Whereas 6-MO *TgAPP^{sw}^{OCN}* mice developed glial activation/brain inflammation selectively in the cortex (see revised Fig. 2), 12-MO *TgAPP^{sw}^{OCN}* mice showed the inflammation phenotype largely in their hippocampus, where OCN-Cre/*APP^{sw}* is weakly expressed in DG hippocampal neurons¹⁶(see Supplementary Figs. 1b and 5). We thus speculate that the hippocampal inflammatory phenotype may be driven by the weak *APP^{sw}*/*A β ₄₂* expression in DG neurons of 12-MO *TgAPP^{sw}^{OCN}* mice. These results were described in Results (pages 6-9) and discussed (pages 16-17).

3. "They examined only in young mice (6-month-old is the oldest). How about these changes in aged *TgAPP^{sw}^{OCN}* mice?"

Response: Good suggestion! As described above, we have examined phenotypes in 12-MO *TgAPP^{sw}^{OCN}* mice. While the cognitive functional deficit is different between 6-MO and 12-MO mutant mice (see revised Fig. 4 and reviewer #2 point 4 above), other phenotypes including brain inflammation and anxiety and depression-like behaviors remain similar or more severe in 12-MO *TgAPP^{sw}^{OCN}* mice (see revised Fig. 3 and supplementary Fig. 5). These results are described in pages 6-10.

4. "What is the mechanism(s) that hippocampal DG neurogenesis is reduced in *TgAPP^{sw}^{OCN}* mice while neuroinflammation in hippocampus is not affected?"

Response: The impaired hippocampal DG neurogenesis in 6-MO *TgAPP^{sw}^{OCN}* mice appeared to be largely due to reduced proliferation of neural stem cells (NSCs) (see revised Supplementary Fig. 6). While multiple reasons may underlie such a deficit, we speculate that the increased SASPs, and reduced growth factors, such as brain-derived neurotrophic factor (BDNF), NGF, insulin-like growth factor 1 (IGF-1), glial cell-line derived neurotrophic factor (GDNF), and basic fibroblast growth factor (bFGF) in *APP^{sw}*⁺ OB-lineage cells may be involved in this deficit for the following reasons. First, NSCs appear to be a group of brain cells highly sensitive to the blood circulation factors, including brain-derived neurotrophic factor (BDNF), nerve growth factor (NGF), insulin-like growth factor 1 (IGF-1), cytokines and chemokines; and bFGF, IGF1 and BDNF play critical roles in promoting neurogenesis¹⁷⁻²³. Second, while multiple cytokines and chemokines are

elevated in APP_{swe} bone cells or serum samples of TgAPP_{swe}^{OCN} mice, BDNF and IGF-1 were reduced in APP_{swe}⁺ BMSCs based on our RNA-seq analysis (see revised Fig. 5d). This point is discussed in revised manuscript (see page 11, 16). We hope to further test this speculation in future experiments.

5. *“Expression level of APP and A β after injection of AAV-CamkII-Cre in LSL-APP_{swe} mice should be assessed.”*

Response: Yes. Done (see revised Supplementary Fig. 8b-g) and described in revised manuscript (see page 10).

Reference

1. Pan, J.-X., *et al.* APP promotes osteoblast survival and bone formation by regulating mitochondrial function and preventing oxidative stress. *Cell death & disease* **9**, 1-18 (2018).
2. Xia, W.F., *et al.* Swedish mutant APP suppresses osteoblast differentiation and causes osteoporotic deficit, which are ameliorated by N-acetyl-L-cysteine. *J Bone Miner Res* **28**, 2122-2135 (2013).
3. Ulland, T.K. & Colonna, M. TREM2 - a key player in microglial biology and Alzheimer disease. *Nature reviews. Neurology* **14**, 667-675 (2018).
4. Paloneva, J., *et al.* DAP12/TREM2 deficiency results in impaired osteoclast differentiation and osteoporotic features. *The Journal of experimental medicine* **198**, 669-675 (2003).
5. Otero, K., *et al.* TREM2 and beta-catenin regulate bone homeostasis by controlling the rate of osteoclastogenesis. *Journal of immunology* **188**, 2612-2621 (2012).
6. Belloy, M.E., Napolioni, V. & Greicius, M.D. A Quarter Century of APOE and Alzheimer's Disease: Progress to Date and the Path Forward. *Neuron* **101**, 820-838 (2019).
7. Zajickova, K., Zofkova, I., Hill, M., Horinek, A. & Novakova, A. Apolipoprotein E 4 allele is associated with low bone density in postmenopausal women. *Journal of endocrinological investigation* **26**, 312-315 (2003).
8. Peter, I., *et al.* Associations of APOE gene polymorphisms with bone mineral density and fracture risk: a meta-analysis. *Osteoporosis international : a journal established as result of cooperation between the European Foundation for Osteoporosis and the National Osteoporosis Foundation of the USA* **22**, 1199-1209 (2011).
9. Cui, S., *et al.* APPswe/Abeta regulation of osteoclast activation and RAGE expression in an age-dependent manner. *J Bone Miner Res* **26**, 1084-1098 (2011).
10. Hsiao, K., *et al.* Correlative memory deficits, Abeta elevation, and amyloid plaques in transgenic mice. *Science* **274**, 99-102 (1996).
11. Oakley, H., *et al.* Intraneuronal beta-amyloid aggregates, neurodegeneration, and neuron loss in transgenic mice with five familial Alzheimer's disease mutations: potential factors in amyloid plaque formation. *The Journal of neuroscience : the official journal of the Society for Neuroscience* **26**, 10129-10140 (2006).
12. Tan, X., *et al.* Smad4 is required for maintaining normal murine postnatal bone homeostasis. *Journal of cell science* **120**, 2162-2170 (2007).
13. Eleftheriou, F. & Yang, X. Genetic mouse models for bone studies--strengths and limitations. *Bone* **49**, 1242-1254 (2011).
14. Kinney, J.W., *et al.* Inflammation as a central mechanism in Alzheimer's disease. *Alzheimer's & dementia* **4**, 575-590 (2018).
15. Akiyama, H., *et al.* Inflammation and Alzheimer's disease. *Neurobiology of aging* **21**, 383-421 (2000).
16. Sun, D., *et al.* Critical Roles of Embryonic Born Dorsal Dentate Granule Neurons for Activity-Dependent Increases in BDNF, Adult Hippocampal Neurogenesis, and Antianxiety-like Behaviors. *Biological psychiatry* (2020).
17. Naghdi, M., Tiraihi, T., Mesbah-Namin, S.A. & Arabkheradmand, J. Induction of bone marrow stromal cells into cholinergic-like cells by nerve growth factor. *Iranian biomedical journal* **13**, 117-123 (2009).
18. Jiang, Y., *et al.* Bone marrow mesenchymal stem cells can improve the motor function of a Huntington's disease rat model. *Neurological research* **33**, 331-337 (2011).
19. Shichinohe, H., *et al.* Bone marrow stromal cells rescue ischemic brain by trophic effects and phenotypic change toward neural cells. *Neurorehabilitation and neural repair* **29**, 80-89 (2015).

20. Zhang, J., *et al.* Expression of insulin-like growth factor 1 and receptor in ischemic rats treated with human marrow stromal cells. *Brain research* **1030**, 19-27 (2004).
21. Cameron, H.A., Hazel, T.G. & McKay, R.D. Regulation of neurogenesis by growth factors and neurotransmitters. *Journal of neurobiology* **36**, 287-306 (1998).
22. Schinkothe, T., Bloch, W. & Schmidt, A. In vitro secreting profile of human mesenchymal stem cells. *Stem cells and development* **17**, 199-206 (2008).
23. Cunningham, C.J., Redondo-Castro, E. & Allan, S.M. The therapeutic potential of the mesenchymal stem cell secretome in ischaemic stroke. *Journal of cerebral blood flow and metabolism : official journal of the International Society of Cerebral Blood Flow and Metabolism* **38**, 1276-1292 (2018).

Reviewers' comments:

Reviewer #1 (Remarks to the Author):

This is an very rigorously designed study, and results are striking and supportive to the conclusion. It is a highly important study, which will gains a broad attentions in the field. Very appropriate for the publication by Communications Biology.

Reviewer #2 (Remarks to the Author):

while i recognize the serious efforts made by the authors, there is a major concern regarding the cascades of events, as detailed in my comments below. Restructuring the manuscript appears as a good possibility but will require additional controls and major revision of the text. Please dont hesitate to contact me for further clarifications.

Reviewer #3 (Remarks to the Author):

I am impressed that the authors have done substantial work to improve the quality of their study in the revised manuscript, and I agree with the idea that senescence phenotype and inflammation in peripheral tissues may exacerbate the neuroinflammation and cognitive alterations in their mouse model. However, I am still not convinced that osteoblastic Swedish mutant APP is related to development of AD. APP transgenic mice with Swedish mutation has been used in AD research because they can recapitulate amyloid pathology in the brain. Basically, it is justified to utilize the APP transgenic mice when the researchers assess amyloid-related alterations in vivo.

As the authors showed in the revised experiments, this effect was only observed in the cell models expressing APP^{swe}, but not APP^{Lon} or APP^{wt}. The age of onset of AD is much younger than sporadic cased in the patients with Swedish mutation. Is there any report of cooccurrence of osteoporosis in early-onset AD, specifically in those with Swedish mutation?

Endoplasmic reticulum (ER) stress can be induced by overexpression of membranous proteins (Hashimoto et al, J Biol Chem 2018). Although Hashimoto et al. did not observe ER stress in single APP transgenic mice, they found that APP and PS1 double transgenic mice showed ER stress in the brain. More importantly, App knock-in mice harboring Swedish mutation did not show ER stress. Considering these facts, I think it's possible that the ER stress in their TgAPP^{swe}OCN mice is induced by overexpression of APP^{swe} protein in osteoblast-lineage cells.

APP^{swe} is reported to alter its metabolism. APP^{swe} can be processed by β -secretase or BACE1 in Golgi-derived vesicles, whereas APP^{wt} is cleaved in endosome after reinternalization and recycling (Haas et al. Nat Med 1995). This alteration in metabolism and cellular localization may explain differences in ER stress specifically induced in APP^{swe} expressed cells. What is the metabolism and cellular localization of APP^{swe}, APP^{Lon}, and APP^{wt} in the osteoblastic cells? Do they also express molecules related to metabolism of APP such as BACE1, presenilin, and ADMA10?

Response to Reviewer #1's comments:

We thank Reviewer 1 for his/her comments that *"This is an very rigorously designed study, and results are striking and supportive to the conclusion. It is a highly important study, which will gain a broad attentions in the field. Very appropriate for the publication by Communications Biology."*

Response to Reviewer #2's comments:

"While i recognize the serious efforts made by the authors, there is a major concern regarding the cascades of events, as detailed in my comments below. Restructuring the manuscript appears as a good possibility but will require additional controls and major revision of the text."

Response: We thank Reviewer 2 for his/her *"concern regarding the cascades of events"*. We appreciate the reviewer's suggestion that *"Restructuring the manuscript appears as a good possibility but will require additional controls and major revision of the text"*. We have carried out additional experiments (e.g., including additional controls, see responses below), and revised the manuscript to incorporate these suggestions (see color highlights in the manuscript).

"Whereas the authors addressed the concerns raised by this reviewer, a key crucial question regarding the cause of brain pathology remains open. At this stage the authors cannot rule out the contribution of brain amyloid, and thus brain inflammation and cognitive decline shown at 12 months of age may indeed reflect the two-hit model of AD, whereby Abeta-induced brain pathology is exacerbated by systemic inflammation."

Response: Agree! We can't rule out the contribution of brain (e.g., dorsal dentate gyrus) amyloid for the hippocampal inflammation and cognitive decline at 12-month-old mutant mice; and our results reflect the two-hit model of AD, and the Abeta-induced brain pathology can be exacerbated by systemic inflammation. The manuscript has thus been largely revised to include these points (see Pages 17-21).

"In addition, it is known that systemic inflammation can cause cognitive decline and some brain inflammation, but this is not unique to osteoblast senescence. As long as the authors cannot rule out the role of brain-endogenous amyloid, the current results seem to better support the two-hit hypothesis than osteoblast-induced AD-like pathology. This is supported by the authors findings that although systemic inflammation is quite robust at 6 months of age, cognitive decline slightly occurred only at 12 months of age when Ab42 accumulates in the brain. Overall, restructuring the manuscript as a model where systemic inflammation potentiates slight amyloid accumulation in the brain to AD-like pathology, appears to better reflect the results. This will require a major revision of the text and ideally, a control group of mice exhibiting similar accumulation of Abeta in the brain in the absence of systemic."

Response: Again, we thank the Reviewer for raising these concerns/suggestions. As responded above, we have revised the manuscript to emphasize the two-hit hypothesis (see Pages 17-18, 21), and to discuss the systemic inflammation's potentiation effect on brain amyloid and AD-like pathology (see Pages 18-19).

However, our studies in TgAPP_{swe}^{OCN} mice suggest the following interesting points. **First**, the systemic inflammation appears to induce anxiety- and depression-like behaviors, but it may be insufficient to cause cognitive decline, because 6-MO TgAPP_{swe}^{OCN} mice exhibit systemic inflammation, cortical brain inflammation, and anxiety- and depression-like behaviors, but not cognitive decline (see revised Figs. 2-4 and 8). **Second**, the cognitive decline appears to be associated with the increase of A β ₄₂ in 12-MO TgAPP_{swe}^{OCN} hippocampus (see revised Fig. 4 and Supplementary Fig. 1b), implicating a role of the weak expression of APP_{swe}/A β ₄₂ in dorsal DG neurons in this deficit. We hope to further test this view in future experiments. **Third**, in terms of the systemic inflammation, while we agree with the reviewer that it can

be induced by deficits in multiple organs, our new results (in revised Fig. 8 and Supplementary Fig. 13 and as described below) lead us to believe that the APP_{swe} induced senescence and SASPs in osteoblast (OB)-lineage cells appear to be a key contributor to this event. **1)** Many (31 over 49, ~63%) upregulated SASP-like factors in serum samples were detected in not only TgAPP_{swe}^{OCN} mice but also Tg2576 mice (a well-characterized AD mouse model expressing APP_{swe} ubiquitously) (see revised Fig. 8f), suggesting that APP_{swe} in OCN-Cre⁺ cells plays a key role in this event. **2)** As pointed out by the reviewer, although APP_{swe} is largely expressed in OB-lineage cells in TgAPP_{swe}^{OCN} mice, we can't rule out the potential contribution to the systemic inflammation by the APP_{swe}'s weak expression in the dorsal DG (dDG) of the hippocampus. To address this concern, we examined the serum inflammatory cytokines and chemokines in mice (LSL-APP_{swe}) injected with AAV-CaMKII-Cre or AAV-GFP into their dDGs. These Cre-injected mice exhibited similar levels of APP_{swe}/Aβ₄₂ in the hippocampus as those of 12-MO TgAPP_{swe}^{OCN} mice (see revised Supplementary Fig. 8g). Using a small-scale antibody array, which contains antibodies against multiple SASP-like pro-inflammatory cytokines and chemokines (Supplementary Fig. 13a), we found that little change (except IL2) between serum samples from the Cre and GFP injected mice (Supplementary Fig. 13a-b). These results thus eliminate the possible contribution to the systematic inflammation by the APP_{swe}/Aβ₄₂ at the dDG, and support the view for APP_{swe} in OCN-Cre⁺ OB-lineage cells to be a major contributor to the systemic inflammation. **3)** Treatments with senescence inhibitors (D+Q) abolished nearly all the increased inflammatory cytokines in serum samples of TgAPP_{swe}^{OCN} mice (see revised Supplementary Fig. 13c-d), supporting the view for senescence and SASPs' contribution to this event.

Finally, it is noteworthy that OB-lineage cells include osteoblasts (OBs), osteocytes, and their precursors, such as bone marrow stromal cells (BMSCs); they come from the bone marrow and are not only key cells for bone formation and bone structure; but also niches for immune cell genesis in the bone marrow, which could produce inflammatory cytokines systemically. Also of interest to note is that bones make up ~15% of a human's total body weight, and in the cortical bone, there are ~ 25,000 osteocytes per mm³ of bone. Thus, a large number of OB-lineage cells is present in the human body, which could make a big impact systemically if they are altered. Together, these observations lead us to believe that APP_{swe} in the OB-lineage cells could be a major contributor to the systemic inflammation.

We have included these points in revised Discussion (see page 17-19).

Response to Reviewer #3's comments:

We thank Reviewer 3 for his/her comments that *"I am impressed that the authors have done substantial work to improve the quality of their study in the revised manuscript, and I agree with the idea that senescence phenotype and inflammation in peripheral tissues may exacerbate the neuroinflammation and cognitive alterations in their mouse model. However, I am still not convinced that osteoblastic Swedish mutant APP is related to development of AD"*.

Response: As responded to Reviewer 2's comments above, we have revised the manuscript to emphasize the two-hit hypothesis (see Pages 17-18, 21), and to discuss the systemic inflammation's potentiation effect on brain amyloid and AD-like pathology (see Pages 18-19).

Specifically, our studies in TgAPP_{swe}^{OCN} mice, as compared with those in Tg2576 mice (a well-studied AD animal model that expresses APP_{swe} ubiquitously), suggest the following points. **First**, the systemic inflammation appears to promote anxiety- and depression-like behaviors, but it is not sufficient to induce the cognitive decline, because 6-MO TgAPP_{swe}^{OCN} mice exhibit systemic inflammation, cortical brain inflammation, and anxiety- and depression-like behaviors, but not cognitive decline (see revised Figs. 2-4 and 8). **Second**, the cognitive decline appears to be associated with the increase of Aβ₄₂ in the TgAPP_{swe}^{OCN} hippocampus (at 12-MO) (see revised Fig. 4 and Supplementary Fig. 1b), suggesting a critical role of APP_{swe}/Aβ₄₂ in dorsal DG neurons for this deficit. We hope to test this view further in future experiments.

Third, in terms of the systemic inflammation, while we agree with the reviewer that it can be induced by deficits in multiple organs, our new results (see below) lead us to believe that APP_{swe} in osteoblast-lineage cells derived senescence and SASPs appear to be a key contributor to this event. **1)** Many (31 over 49, ~63%) upregulated SASP-like factors were detected in serum samples of not only TgAPP_{swe}^{OCN} mice but also Tg2576 mice (see revised Fig. 8f), suggesting that APP_{swe} in OCN-Cre⁺ cells plays a key role in this event. **2)** As pointed out by the reviewer, although APP_{swe} is largely expressed in OB-lineage cells in TgAPP_{swe}^{OCN} mice, we can't rule out the potential contribution to the systemic inflammation by the APP_{swe}'s weak expression in the dorsal DG (dDG) of the hippocampus. To address this concern, we examined the serum inflammatory cytokines and chemokines in mice (LSL-APP_{swe}) injected with AAV-CaMKII-Cre or AAV-GFP into their dDGs. These Cre-injected mice exhibited similar levels of APP_{swe}/A β 42 in the hippocampus as those of 12-MO TgAPP_{swe}^{OCN} mice (see revised Supplementary Fig. 8g). Using a small-scale antibody array, which contains antibodies against multiple SASP-like pro-inflammatory cytokines and chemokines (Supplementary Fig. 13a), we found that little to no change between serum samples from the Cre and GFP injected mice (Supplementary Fig. 13a-b). These results thus eliminate the possible contribution to the systematic inflammation by the APP_{swe}/A β 42 at the dDG, and support the view that APP_{swe} in OCN-Cre⁺ OB-lineage cells is likely to be a major contributor to the systemic inflammation. **3)** Treatments with senescence inhibitors (D+Q) abolished nearly all the increased inflammatory cytokines in serum samples of TgAPP_{swe}^{OCN} mice (see revised Supplementary Fig. 13c-d), supporting the view for senescence and SASPs' contribution to this event.

Finally, it is noteworthy that osteoblast-lineage cells include osteoblasts (OBs), osteocytes (Ocys), and their precursors, such as bone marrow stromal cells (BMSCs); they come from the bone marrow and are the key cells for bone formation and bone structure; and bone cells and bone marrow cells are tightly intertwined, and osteoblast-lineage cells play crucial roles in regulating immune cell production and functions in producing inflammatory cytokines systemically. Also of interest to note is that bones make up ~15% of a person's total human body weight, and in the cortical bone, there are ~ 25,000 osteocytes per mm³ of bone. Thus, a large number of osteoblast-lineage cells is present in the human body, which could make a big impact systemically if they are altered. Together, these observations lead us to believe that APP_{swe} in OB-lineage cells is a major contributor to the systemic inflammation.

We have included these points in revised Discussion (see pages 17-19).

“APP transgenic mice with Swedish mutation has been used in AD research because they can recapitulate amyloid pathology in the brain. Basically, it is justified to utilize the APP transgenic mice when the researchers assess amyloid-related alterations in vivo. As the authors showed in the revised experiments, this effect was only observed in the cell models expressing APP_{swe}, but not APP_{Lon} or APP_{wt}. The age of onset of AD is much younger than sporadic cases in the patients with Swedish mutation. Is there any report of cooccurrence of osteoporosis in early-onset AD, specifically in those with Swedish mutation?”

Response: This is an interest point! Although it is reported that bone mineral density is reduced in the earliest clinical stages of AD patients (both men and women) and associated with their brain atrophy and memory decline (see Loskutova et al., J. Alzheimer's Dis., 2009), it remains unclear if the AD patients carrying the Swedish mutation have osteoporosis-like deficit. We hope that our study will stimulate investigators to address this question in future.

“Endoplasmic reticulum (ER) stress can be induced by overexpression of membranous proteins (Hashimoto et al, J Biol Chem 2018). Although Hashimoto et al. did not observe ER stress in single APP transgenic mice, they found that APP and PS1 double transgenic mice showed ER stress in the brain. More importantly, App knock-in mice harboring Swedish mutation did not show ER stress. Considering these facts, I think it's

possible that the ER stress in their TgAPP^{swe}OCN mice is induced by overexpression of APP^{swe} protein in osteoblast-lineage cells.”

Response: We thank the Reviewer for providing these insights. To address if the ER stress in OB-lineage cells is induced by over expression of APP_{swe}, we compared ER stress (by GRP78, a sensor for ER stress) in MC3T3 cells expressing APP_{swe}-YFP with those in APP_{wt}-YFP or APP_{lon}-YFP expressing MC3T3 cells. Interestingly, MC3T3 cells expressing APP_{swe}-YFP, but not APP_{wt}-YFP or APP_{lon}-YFP, showed an increase in GRP78 (see revised Supplementary Fig. 14a-b). In addition, APP_{swe}-YFP had a more prominent co-localization with GRP78 than those of APP_{wt}-YFP or APP_{lon}-YFP (see revised Supplementary Fig. 14a, c). These results argue for the specificity of the induction of ER stress by expression of APP_{swe}-YFP, rather than by expression of any membrane protein, in line with the view for APP_{swe} to induce cellular senescence likely by ER-stress.

In terms of the reports that little ER stress is detected in single APP transgenic mice or App knock-in mice harboring Swedish mutation (Hashimoto et al, J Biol Chem 2018), we speculate that APP_{swe} induction of ER stress may be cell type/tissue specific; and OB-lineage cells may be more sensitive to APP_{swe} than neurons in its induction of ER stress. We have included these points in revised manuscript (see Pages 19-20).

“APP^{swe} is reported to alter its metabolism. APP^{swe} can be processed by β -secretase or BACE1 in Golgi-derived vesicles, whereas APP^{wt} is cleaved in endosome after reinternalization and recycling (Haas et al. Nat Med 1995). This alteration in metabolism and cellular localization may explain differences in ER stress specifically induced in APP^{swe} expressed cells. What is the metabolism and cellular localization of APP^{swe}, APP^{lon}, and APP^{wt} in the osteoblastic cells? Do they also express molecules related to metabolism of APP such as BACE1, presenilin, and ADMA10?”

Response: Again, we thank the Reviewer for raising these questions/suggestions. As described above, we have examined the cellular localizations of APP_{swe}-YFP, APP_{lon}-YFP, and APP_{wt}-YFP in MC3T3 cells, an osteoblastic cell line, as suggested. Interestingly, APP_{swe}-YFP exhibited distinctive features not only in its activation of ER stress, but also in its subcellular localizations, from those of APP_{wt}-YFP or APP_{lon}-YFP. APP_{swe}-YFP had an increased co-localization with EEA1, an early endosome marker, but a decreased co-location with GM130, a marker for Trans-Golgi, as compared with those of APP_{wt}-YFP or APP_{lon}-YFP. These results thus reveal a potential cellular mechanism underlying the selective induction of the ER stress and senescence by APP_{swe}, but not by APP_{wt} or APP_{lon}. These results are also in line with “the view for APP_{swe} to be processed by β -secretase or BACE1 in Golgi-derived vesicles, and APP_{wt} to be cleaved in endosomes (Haas et al. Nat Med 1995)”. We have included these results in revised Supplementary Fig. 14a-g, and described in Pages 15-16, 19-20.

We also examined BACE1, presenilin, and ADMA10's expression in OB-lineage cells as suggested. The expression levels of BACE1, presenilin, and ADAM10 in OB-lineage cells appeared lower than those in brain (cortex and hippocampus) (see Fig. below). These results thus provide an explanation for much lower levels of A β _{40/42} were detected in OB-lineage cells, where APP_{swe} was largely expressed in these cells in TgAPP_{swe}^{OCN} mice. These results are shown here for your reference, but not in revised manuscript, because the current manuscript contains 10 Figures, and 14 supplementary Figures (reached the maximum numbers of Figures that the Journal is allowed), and these data are not directly relevant to the current experimental objectives.

Figure. The expression levels of BACE1, presenilin, and ADAM10 in OB-lineage cells (a) Western blot analysis using indicated antibodies in lysates of mouse cortex, hippocampus, MC3T3 and BMSCs. (b) RT-PCR analysis of *Bace1*, *Adam10* and *Psen1* gene expression in cortex, hippocampus, MC3T3 and BMSCs. The data were presented as mean \pm SD, $n=3$, $*p<0.05$, by Mann-Whitney U test. **Note:** lower expression in *Bace1* in OB-lineage cells (MC3T3 and BMSCs) than those in the brain (mouse cortex and hippocampus) was detected.

REVIEWERS' COMMENTS:

Reviewer #2 (Remarks to the Author):

The authors addressed all the critics raised in my recent review. I thus approve the manuscript for publication.

Minor comment: the manuscript will benefit from further language and grammar editing.

Reviewer #3 (Remarks to the Author):

I think most of my concern is appropriately addressed in their revised manuscript. I just would like the authors to include a description about human data in the discussion section such as their reply in rebuttal as follows:

Although it is reported that bone mineral density is reduced in the earliest clinical stages of AD patients (both men and women) and associated with their brain atrophy and memory decline (see Loskutova et al., J. Alzheimer's Dis., 2009), it remains unclear if the AD patients carrying the Swedish mutation have osteoporosis-like deficit. We hope that our study will stimulate investigators to address this question in future.